# Bridging the gap between $f$-divergences and Bayes Hilbert spaces

**Linus Lach**[*]**, Alexander Fottner, Yarema Okhrin**
Department of Statistics and Data Science
University of Augsburg
{linus.lach,alexander.fottner,yarema.okhrin}@uni-a.de

## Abstract

We introduce a novel framework that generalizes $f$-divergences by incorporating divergence-generating functions that exhibit non-convex behavior in a neighborhood of the origin. Using this extension, we define a new class of pseudo $f$-divergences, encompassing a wider range of distances between distributions that traditional $f$-divergences cannot capture. Among these, we focus on a particular pseudo divergence, obtained by considering the induced metric of Bayes Hilbert spaces. Bayes Hilbert spaces are frequently used due to their inherent connection to Bayes's theorem as they allow sampling from potentially intractable posterior densities, a challenging task until now. In the more general context, we prove that pseudo $f$-divergences are well-defined and introduce a variational estimation framework that can be used in a statistical learning context. By applying this variational estimation framework to $f$-GANs, we achieve improved FID scores over existing $f$-GAN architectures and competitive results with the Wasserstein GAN, highlighting its potential for both theoretical research and practical applications in learning theory.

## 1 Introduction

A central challenge in learning theory is estimating statistical divergences between distributions from finite samples to quantify their dissimilarity. $f$-divergences are a popular class of divergences (Csiszár et al., 2004; Liese & Vajda, 2006) including the Kullback-Leibler (KL) and Pearson divergence. Their general form is given by $D_f(\mu, \nu) = \int f(\frac{d\nu}{d\mu})d\mu$, where $f$ is a convex lower-semicontinuous function satisfying $f(1) = 0$. $f$-divergences and the subsequent estimation have been extensively studied in Nguyen et al. (2010); Rubenstein et al. (2019); Nguyen et al. (2009); Moon & Hero (2014); Keziou (2003). However, research on generalizing these divergences (Gimenez & Zou, 2022; Birrell et al., 2022; Chen et al., 2024), particularly by relaxing the convexity constraint, still needs to be explored.

We introduce a novel pseudo $f$-divergence framework that relaxes the convexity constraint on divergence-generating functions, offering greater flexibility and applicability in various machine-learning tasks. This flexibility is crucial in tasks like generative image modeling (Nowozin et al., 2016), or topic modeling on large text data sets (Hoffman et al., 2013), where the underlying data is diverse. Previously, the convexity requirement hindered connections between $f$-divergences and other dissimilarity measures like the Bayes Hilbert space metric (Van den Boogaart et al., 2014), frequently used in Bayesian statistics. In Bayesian statistics, inference typically requires sampling from intractable densities (Kroese et al., 2013). Sampling from intractable densities using divergences in a bayesian setting has been studied in Blei et al. (2017); Campbell & Broderick (2018); Campbell & Beronov (2019). However, these methods have only been applied using the KL-divergence (Blei et al., 2017; Campbell & Beronov, 2019) or lower dimensional synthetic data sets (Campbell & Broderick, 2018).

To highlight the application of general $f$-divergences in a Bayesian statistics context, we therefore explore a connection between $f$-divergences and Bayes Hilbert spaces, which provide an alternative structure for measuring distances between distributions. Bayes Hilbert spaces are distribution spaces with a vector structure, intrinsically connected to Bayes's theorem, where addition aligns with

Bayesian updating (Van den Boogaart et al., 2014). Due to their Hilbert space structure, they are equipped with an induced metric, providing a natural notion of distance between measures that can be incorporated into the proposed pseudo $f$-divergence framework. Another advantage of estimating the Bayes Hilbert space metric over traditional divergences arises from the Hilbert space structure. The Hilbert space structure implies that every Cauchy sequence converges within the space, which can be exploited in an optimization setting. By modeling the loss function as a sequence in a Bayes Hilbert space and using gradient-based approaches, convergence to a solution near the optimum is ensured, preventing instability near the optimum. However, defining a Hilbert space for probability measures is not straightforward (Van den Boogaart et al., 2010). First, probability measures must be identified as vectors before imposing a linear structure on them to obtain a linear space. This linear space can then be endowed with an inner product to define a Hilbert space. The linear space structure allows probability measures and their associated densities to be embedded in a subspace of $L^2$, enabling a clear understanding of the distances between distributions.

The connection between $f$-divergences and Bayes Hilbert spaces can be established using the clr transformation for distributions, which is defined as $\text{clr}(\mu) = \log(\frac{\mathrm{d}\mu}{\mathrm{d}\lambda}) - \int \log(\frac{\mathrm{d}\mu}{\mathrm{d}\lambda})\mathrm{d}\lambda$, where $\frac{\mathrm{d}\mu}{\mathrm{d}\lambda}$ is the Radon-Nikodym derivative of $\mu$ with respect to some continuous base measure $\lambda$. The induced metric of a Bayes Hilbert space, which involves the squared clr transformation, closely relates to a pseudo $f$-divergence with the pseudo-divergence-generating function $x \mapsto x \log(x)^2$. Since this function encompasses a local non-convexity on the interval $[0, \exp(-1)]$, we apply our newly developed framework, which allows us to estimate the so-called BHS divergence and also exploit the characteristics of Bayes Hilbert spaces to obtain a lower bound for the pseudo divergence that is attained.

## 1.1 Contributions

Our main contributions can be summarized as follows:

- We develop a novel theoretical framework bridging the gap between $f$-divergences and Bayes Hilbert spaces.
- We generalize $f$-divergences by allowing for locally non-convex divergence-generating functions.
- We present an efficient sampling algorithm for high dimensional distributions.
- We apply our extended pseudo-divergence estimation framework to $f$-GANs, outperforming traditional $f$-GANs achieving lower FID scores.

## 1.2 Related Work

$f$-**divergences:** Our work extends the existing body of research on $f$-divergences and their application in learning theory, particularly in the context of sampling from unknown distributions. A foundational approach to estimating $f$-divergences was developed by Nowozin et al. (2016); Nguyen et al. (2010), where the authors developed and utilized a variational estimation framework to estimate $f$-divergences with an application to GANs. Their framework allows for flexible divergence estimation but remains constrained by the convexity of divergence-generating functions. Our approach addresses this limitation by relaxing the convexity constraint, allowing for a broader range of pseudo $f$-divergences that can be tailored to specific tasks.

Besides $f$-divergences, Rényi divergences, which generalize the KL divergence by introducing a parameter $\alpha > 0, \alpha \neq 1$ that corresponds to a weight controlling the sensitivity of divergence regarding areas where the distributions are similar or dissimilar have been developed to provide greater flexibility (Van Erven & Harremos, 2014). While Rényi divergences provide greater flexibility compared to KL, they are still limited by the convexity constraint inherent in $f$-divergences.

A different approach to measure the dissimilarity between probability measures is offered by integral probability metric (IPM) such as the Wasserstein distance. The Wasserstein distance has gained popularity in generative modeling due to its robustness in comparing distributions with different supports (Sriperumbudur et al., 2012). However, as Yatracos (2022) has noted, the Wasserstein distance is unsuitable for estimating distributions with heavy tails, as it struggles to handle distributions whose tails are not exponentially bounded. Recent work by Chen et al. (2024) addresses this issue by

proposing a penalized version of the Wasserstein distance, which combines it with $\alpha$-divergences to estimate heavy-tailed distributions better. While promising, their method requires careful tuning of the penalty terms and relies on the Wasserstein metric's limitations, particularly in high-dimensional spaces.

In contrast, Birrell et al. (2022) combines IPMs and $f$-divergences introducing $(f, \Gamma)$-divergences that inherit the ability to compare distributions that are not absolutely continuous from IPMs, while also retaining the property of $f$-divergences to control heavy-tailed distributions. Although this hybrid approach broadens the applicability of divergence estimation, it still imposes the convexity constraint on the divergence-generating function.

**Bayes Hilbert Spaces:** Bayes spaces, and their extension to Bayes Hilbert spaces (Van den Boogaart et al., 2010; 2014; Egozcue et al., 2006) while still in their infancy, were introduced as a natural generalization of the theory of compositional data, originally popularized by Aitchison in Aitchison (1982; 1986); Van den Boogaart & Tolosana-Delgado (2013). These spaces address the challenge of defining meaningful distances between probability measures for specific data analysis tasks. The core idea is to introduce a vector space structure for measures, where the group operations align with Bayes updating (Van den Boogaart et al., 2014), thus creating a framework in which Bayesian reasoning can be incorporated directly into the space of measures.

Bayes Hilbert spaces are frequently used in functional regression, including Maier et al. (2021); Arata (2017); Machalova et al. (2016); Machalová et al. (2021); Talská et al. (2018), where Radon-Nikodym derivatives are modeled using spline functions within the Bayes Hilbert space.

However, one significant challenge with these approaches is their reliance on functional regression, where the targets or features are typically density functions. Since these densities are not directly observed in practice, the models must infer them from finite samples generated by the underlying distributions. This creates a gap between the theoretical framework of Bayes Hilbert spaces and practical learning scenarios.

Our work addresses this gap by incorporating a generative adversarial network (GAN) to approximate the underlying probability distribution while simultaneously minimizing the distance between the learned and true density in the Bayes Hilbert space. This integration allows us to bridge the functional models of Bayes spaces with finite-sample learning, offering a more practical and scalable solution to regression tasks involving density functions.

In parallel, Wynne (Wynne, 2023) explored the use of Bayes Hilbert spaces for Bayesian posterior approximation, concluding that these spaces provide a natural and appropriate framework for approximating posterior distributions. Our approach builds on this insight by extending its application to finite sample learning and divergence minimization in the Bayes Hilbert space.

## 1.3 OUTLINE

We begin by reviewing key properties of $f$-divergences and establish theoretical assumptions on the underlying spaces. Next, we address challenges in quantifying dissimilarities between measures using $f$-divergences. Since estimating $f$-divergences relies on a variational framework involving Fenchel conjugates, we discuss how these conjugates are used in a dual representation of convex divergence-generating functions to derive an estimable lower bound. We then outline the conditions for realizing this bound and present its analytical form, which remains valid under milder assumptions.

In Section 3, we review Bayes Hilbert spaces as a framework for measuring distances between probability measures by embedding them into function spaces with a separable Hilbert space structure defined through the clr. The induced metric corresponds to the log-variance, which connects to $f$-divergences.

Finally, we apply this generalized divergence estimation framework to $f$-GANs, demonstrating that the more flexible pseudo-divergence approach outperforms traditional $f$-divergences.

Proofs of our theoretical results are provided in the Appendix.

## 2 $f$-DIVERGENCES

Statistical divergences measure the dissimilarity between probability distributions and are often used to estimate parameters in statistical models, such as generative algorithms designed to sample from unknown distributions. $f$-divergences are a widely used class of these measures defined as:

$$D_f(\mu, \nu) = \int_\Omega p_\nu(x) f\left(\frac{p_\mu(x)}{p_\nu(x)}\right) \lambda(\mathrm{d}x).$$

where $\mu$ and $\nu$ are assumed to be two probability distributions on the same measure space $(\Omega, \mathcal{B}, \lambda)$ ($\mathcal{B}$ denoting the Borel $\sigma$-field over $\Omega$) that are both absolutely continuous with respect to the $\sigma$-finite base measure $\lambda$. $p_\mu$ and $p_\nu$ denote the Radon-Nikodym densities of $\mu$ and $\nu$ with respect to $\lambda$. Common choices for $\lambda$ are the Lebesgue measure or counting measure. $f$ is an arbitrary convex, lower-semicontinuous function $f : \mathbb{R}_+ \to \mathbb{R}$, satisfying $f(1) = 0$ called divergence generating function. By requiring that $f(1) = 0$, it is ensured that if $\mu = \nu$ a.s., $D_f(\mu, \nu) = 0$ and the lower-semicontinuity of $f$ ensures that $\lim_{t \searrow 1} f(t) = f(0)$. Note, that $f$-divergences generally do not induce metrics as they are not symmetric in most cases, i.e., $D_f(\mu, \nu) \neq D_f(\nu, \mu)$. However, they can still effectively be used for measuring dissimilarity or pseudo distances between measures. Since the function $f$ must be convex, which can be restrictive when defining new divergences, it potentially limits the range of distributions a generative model can learn effectively. A challenge with $f$-divergences is their difficulty in estimation from finite samples of $\mu$ and $\nu$. To address this, particularly in estimating likelihood ratios for multivariate distributions, convex optimization techniques are employed (Nguyen et al., 2010). The optimization uses variational estimation involving the Fenchel conjugate of a continuous convex function defined as:

$$f^*(y) := \sup_{x \in \mathrm{dom}(f)} \{x^T y - f(x)\}, \quad y \in \mathrm{dom}(f^*).$$

This implies the dual representation of $f$, known as the Fenchel or convex duality:

**Theorem 2.1 (Rockafellar, 1970).** *Let $f : \mathbb{R}^n \to \mathbb{R}$ be a convex function. If $f$ is lower semi-continuous, then the duality $f^{**}(x) = f(x)$ for all $x \in \mathbb{R}^n$ holds.*

Leveraging the dual representation $f(x) = f^{**}(x) = \sup_{y \in \mathrm{dom}(f^*)}\{x^T y - f^*(y)\}$ in the sense of Theorem 2.1, a lower bound for the divergence of $\mu$ and $\nu$ can be derived (Nguyen et al., 2010):

$$D_f(\mu, \nu) \geq \sup_{T \in \mathcal{T}} \left\{ \mathbb{E}_\mu(T) - \mathbb{E}_\nu(f^* \circ T) \right\}. \tag{1}$$

Here, $\mathcal{T}$ denotes an arbitrary class of Borel measurable functions $T : \Omega \to \mathrm{dom}(f^*)$. By adding the additional assumption that $f$ is continuously differentiable on its entire domain, the bound in equation 1 is tight and the supremum can be represented analytically as

$$\tilde{T}(x) := f'\left(\frac{p_\mu(x)}{p_\nu(x)}\right).$$

By adapting the definition of convex conjugates to allow concave functions instead of convex functions, Theorem 2.1 can be adjusted to hold for upper-semicontinuous, concave functions as well. The concave conjugate of a function $f : \mathbb{R}^n \to \mathbb{R}$ is defined as

$$f_*(y) := \inf_{x \in \mathrm{dom}(f)} \{x^T y - f(x)\}, \quad y \in \mathrm{dom}(f_*).$$

Fenchels duality in the concave case then states $f_{**}(x) = f(x)$ for all $x \in \mathbb{R}^n$ (Rockafellar, 1970).

## 3 BAYES HILBERT SPACES

Another idea for quantifying distances between probability measures is the embedding into suitable spaces of functions such as Hilbert spaces. The induced metric of these spaces can then be used to measure the distance between those probability measures. This basic idea will be formulated mathematically in the following paragraph which is based on Van den Boogaart et al. (2014; 2010).

Denote by $\mathcal{M}$ the set of $\sigma$–finite measures on some Borel measurable space $(\Omega, \mathcal{B})$ that are equivalent to a $\sigma$–finite base measure $\lambda$ on $\mathcal{B}$. Then, $\mu$ and $\nu$ are called $B$–equivalent, denoted by $\mu =_{B(\lambda)} \nu$, if there exists a constant $c > 0$ such that $\mu(A) = c\nu(A)$ for all $A \in \mathcal{B}$ with the convention $c \cdot (+\infty) = +\infty$. Define $B(\lambda)$ as the set of $=_B$–equivalent classes, i.e., the quotient space $\mathcal{M}(\lambda)/ =_B$. The relation $=_B$ is an equivalence relation on $\mathcal{M}(\lambda)$. Since the considered measures are equivalent to each other, the Radon–Nikodym theorem ensures that the densities $d\mu/d\nu$ exist for every $\mu, \nu \in \mathcal{M}(\lambda)$.

Given the equivalence class of $=_B$–equivalent measures, addition and multiplication can be defined: For $R \in \mathcal{B}$ define the perturbation of $\mu$ by $\nu$, the powering of $\mu$ for a scalar $\alpha \in \mathbb{R}$, and the negative perturbation in $B(\lambda)$ as

$$(\mu \oplus \nu)(R) := \int_R \frac{d\mu}{d\lambda}\frac{d\nu}{d\lambda}\, d\lambda, \quad (\alpha \odot \mu)(R) := \int_R \left(\frac{d\mu}{d\lambda}\right)^\alpha d\lambda, \quad (\mu \ominus \nu)(R) := \int_R \frac{d\mu}{d\nu}\, d\lambda.$$

In Bayes Hilbert space literature it is common to identify measures with their corresponding Radon–Nikodym densities. This is coherent in the sense that by setting $\mu =_B \frac{d\mu}{d\lambda}$ for every $\mu \in B(\lambda)$ it holds that $\mu \oplus \nu =_B \frac{d\mu}{d\lambda}\frac{d\nu}{d\lambda}$.

$(B(\lambda), \oplus, \odot)$ is a real linear space called Bayes linear space with base measure $\lambda$. The additive neutral element is given by $0_B := \lambda$ and the additive inverse element $\ominus\mu := \frac{d\lambda}{d\mu}$. To derive a Hilbert space based on this vector space, we now assume that the base measure $\lambda$ is a probability measure. The space $B^p(\lambda)$, $p \geq 1$ of $\lambda$–equivalent measures defined as

$$B^p(\lambda) := \left\{\mu \in B(\lambda) : \int \left|\log\left(\frac{d\mu}{d\lambda}\right)\right|^p d\lambda < +\infty\right\}$$

is a vector subspace of $B(\lambda)$. For details and more basic properties, see Van den Boogaart et al. (2010).

Let $\mu \in B^1(\lambda)$ with corresponding density $\frac{d\mu}{d\lambda}$. Then, define the centered log-ratio transformation clr of $\mu$ as

$$\mathrm{clr}(\mu) := \log\left(\frac{d\mu}{d\lambda}\right) - \int \log\left(\frac{d\mu}{d\lambda}\right) d\lambda.$$

The clr transformation has the desirable property of maintaining the relation of $\oplus$ with multiplication and $\odot$ with exponentiation respectively which is enforced by the properties of the logarithm. Furthermore, the map $\mathrm{clr} : B^1(\lambda) \to L_0^1(\lambda)$ is linear.

Recall, that $L_0^1(\lambda) = \{g \in L^1(\lambda) : \int g\, d\lambda = 0\}$. Given the map clr, an inner product on $B^2$ can be defined which in turn induces a metric.

Let therefore $p_\mu$ and $p_\nu$ be two Radon-Nikodym densities in $B^2(\lambda)$, i.e., for $\mu, \nu \in B^2(\lambda)$ define $p_\mu(x) := \frac{d\mu}{d\lambda}(x)$ and $p_\nu(x) := \frac{d\nu}{d\lambda}(x)$ for all $x \in \mathbb{R}^n$. Then, the inner product $\langle p_\mu(x), p_\nu(x)\rangle_{B^2(\lambda)}$ is defined as:

$$\int_\Omega \left(\log(p_\mu(x)) - \int \log(p_\mu(y))\, \lambda(dy)\right)\left(\log(p_\nu(x)) - \int \log(p_\nu(y))\, \lambda(dy)\right) \lambda(dx).$$

$B^2(\lambda)$ is a separable Hilbert space and the map $\mathrm{clr} : B^2(\lambda) \to L_0^2(\lambda)$ is an isometry of Hilbert spaces.

The map $d_{B^2(\lambda)} : B^2(\lambda) \times B^2(\lambda) \to [0, \infty)$, $d_{B^2(\lambda)}(p_\mu, p_\nu) := \left(\langle p_\mu \ominus p_\nu, p_\mu \ominus p_\nu\rangle_{B^2(\lambda)}\right)^{\frac{1}{2}}$ is an induced metric and the pair $(B^2(\lambda), d_{B^2(\lambda)})$ is a complete metric space.

If we want to construct an $f$-divergence that is closely related to the aforementioned metric, its explicit form is of major interest. We can derive such an analytical expression of the Bayes Hilbert space metric using the following Lemma.

**Lemma 3.1.** *Let $p_\mu$ and $p_\nu$ be two Radon–Nikodym densities in $B^2(\lambda)$. Then,*

$$d_{B^2(\lambda)}(p_\mu, p_\nu)^2 = \mathrm{Var}_\lambda(\log(\mu \ominus \nu)).$$

Therefore the Bayes Hilbert space metric directly corresponds to the log-variance loss studied in e.g. Richter et al. (2020); Nüsken & Richter (2021); Richter & Berner (2024), embedding those works in the area of Bayes Hilbert space learning.

# 4 BRIDGING THE GAP BETWEEN BAYES HILBERT SPACES AND $f$-DIVERGENCES

In this section, we show how a pseudo $f$-divergence can be formulated to directly relate to the Bayes Hilbert space metric $d_{B^2}$, enabling sampling from a probability distribution within a well-defined Bayes Hilbert space. This approach addresses a key challenge in Bayes Hilbert space learning, as highlighted in Maier et al. (2021): estimating densities through direct sampling from the learned distribution.

## 4.1 MIXED CONJUGATES

A key challenge is that $f$-divergences are only well-defined when $f$ is lower semi-continuous and convex. Thus to establish the desired connection, we must first extend the definition of $f$-divergences to handle local subsets near the origin where $f$ is not convex. Consider a continuous function $f : [0, \infty) \to \mathbb{R}$, which is concave on $[0, a)$ for some $a > 0$ and convex on $[a, \infty)$. To account for this local non-convexity, we define the mixed conjugate $f_*^*$ of $f$ as

$$f_*^*(t) := \sup_{x \in [a, \infty)} \{tx - f(x)\} \mathbb{1}_{\{t \in M\}} + \inf_{x \in [0, a)} \{tx - f(x)\} \mathbb{1}_{\{t \in N\}}, \tag{2}$$

where

$$M := \big\{ t \in \mathrm{dom}(f_*^*) : \underset{x \in \mathrm{dom}(f)}{\mathrm{argmax}} (tx - f(x)) \in [a, \infty) \big\},$$

$$N := \big\{ t \in \mathrm{dom}(f_*^*) : \underset{x \in \mathrm{dom}(f)}{\mathrm{argmin}} (tx - f(x)) \in [0, a) \big\}.$$

The set $M$ ($N$) ensures that we consider the convex (concave) conjugate on the domain of $f_*^*$ where $f$ is convex (concave). Under mild conditions, this definition allows a disjoint characterization of the subdomains $M$ and $N$. The motivation behind this mixed conjugate is straightforward: it enables a well-defined dual representation for locally non-convex functions.

**Lemma 4.1.** *For any continuous function $f : [0, \infty) \to \mathbb{R}$ that is convex on $[a, \infty)$, $a > 0$, concave on $[0, a)$, and satisfies $\lim_{x \to \infty} f(x) = +\infty$, the mixed conjugate satisfies:*

1. *$M \cap N = \emptyset$ and $M \cup N = \mathrm{dom}(f_*^*)$*

2. *$f_{**}^{**} = f$ for all $t \in \mathrm{dom}(f)$, where $f_{**}^{**}$ denotes the biconjugate of $f$, i.e. $f_{**}^{**} = (f_*^*)_*^*$.*

The unboundedness assumption on $f$ is not only a theoretical necessity in the proof of the lemma above but also carries significance in the context of divergence-generating functions. By assuming that $f$ is unbounded, we permit the dissimilarity between probability measures to grow arbitrarily large.

At first glance, it is not clear why such pseudo-divergences are well-defined, however, the following Lemma proves their coherence.

**Lemma 4.2.** *Let $f : [0, \infty) \to [0, \infty)$ be concave on some interval $[0, a)$, $a > 0$ and convex on $[a, \infty)$. For any probability measures $\mu, \nu \in B^2(\lambda)$, $D_f(\mu, \nu)$ is well-defined in the sense that $D_f(\mu, \nu) \geq 0$ and $D_f(\mu, \nu) = 0 \iff \mu = \nu \quad \lambda - a.s.$*

To address the challenge of estimating likelihood ratios from finite samples, we build on the approach in Nguyen et al. (2010) by developing a generalized variational framework applicable to pseudo $f$-divergences.

**Theorem 4.3.** *Let $f : [0, \infty) \to \mathbb{R}$, convex on $[a, \infty)$ with $0 < a < \infty$, concave on $[0, a)$, assume that $f$ is twice continuously differentiable, and that $((f_*^*)')^{-1}$ is $\lambda$-a.s. well-defined. Furthermore, let $\mu, \nu \in B^2(\lambda)$ with $\frac{\mathrm{d}\mu}{\mathrm{d}\lambda}(x) := p_\mu(x)$, $\frac{\mathrm{d}\nu}{\mathrm{d}\lambda}(x) := p_\nu(x)$, $x \in \mathbb{R}$. Then, $\tilde{T}(x) := f'\left(\frac{p_\mu(x)}{p_\nu(x)}\right)$ is an optimizer for*

$$\sup_{T \in C(M^*)} \big\{ \left( \mathbb{E}_\mu(T) - \mathbb{E}_\nu(f_*^* \circ T) \right) \big\} + \inf_{T \in C(N^*)} \big\{ \left( \mathbb{E}_\mu(T) - \mathbb{E}_\nu(f_*^* \circ T) \right) \big\}. \tag{3}$$

*Here, $C(M^*)$ ($C(N^*)$) denotes the set of continuous functions $T : \Omega \to \mathrm{dom}(f_*^*)$ such that $f_*^* \circ T$ is convex (concave).*

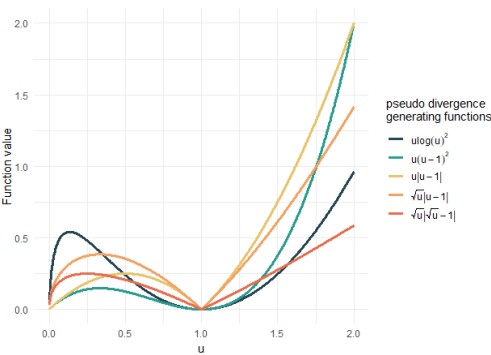

Figure 1: Plot of different pseudo-divergence-generating functions $f$

Table 1: Table of different pseudo-divergence-generating functions and their respective pseudo divergence

| $f(u)$ | $D_f(\mu, \nu)$ |
|---|---|
| $u \log(u)^2$ | $\mathbb{E}_\mu \left( \log \left( \frac{p_\mu(x)}{p_\nu(x)} \right)^2 \right)$ |
| $u(u-1)^2$ | $\mathbb{E}_\mu \left( \left( \frac{p_\mu(x)}{p_\nu(x)} - 1 \right)^2 \right)$ |
| $u\|u-1\|$ | $\mathbb{E}_\mu \left( \left\| \frac{p_\mu(x)}{p_\nu(x)} - 1 \right\| \right)$ |
| $\sqrt{u}\|u-1\|$ | $\mathbb{E}_\mu \left( \left\| \frac{\sqrt{p_\mu(x)}}{\sqrt{p_\nu(x)}} - \frac{\sqrt{p_\nu(x)}}{\sqrt{p_\mu(x)}} \right\| \right)$ |
| $\sqrt{u}\|\sqrt{u}-1\|$ | $\mathbb{E}_\mu \left( \left\| 1 - \frac{\sqrt{p_\nu(x)}}{\sqrt{p_\mu(x)}} \right\| \right)$ |

Theorem 4.3 can be applied to the extended class of divergence-generating functions. A brief selection of such pseudo-divergence-generating functions and their respective pseudo-divergence can be in Figure 1 and Table 1.

The optimization problem in equation 3 can be transformed to reduce the two objectives into a single equivalent objective with the same optimum. This transformation leverages the property $\inf(\cdot) = -\sup(\cdot)$ and substituting $T$ with $\bar{T} = T(\mathbb{1}_{\{T \in C(M^*)\}} - \mathbb{1}_{\{T \in C(N^*)\}})$. For details, see Proof of Corollary A.1.

**Corollary 4.4.** *For any function $f$ satisfying the conditions of Theorem 4.3*

$$D_f(\mu, \nu) \geq \sup_{\bar{T} \in C(M^*) \cup C(N^*)} \left\{ \mathbb{E}_\mu(\bar{T}) - \mathbb{E}_\nu(f_*^* \circ \bar{T}) \right\}. \tag{4}$$

Therefore, by applying the generalizing mixed conjugate to the variational estimation framework of Nguyen et al. (2010), we can still derive a lower bound for the pseudo $f$–divergence.

## 4.2 Bayes Hilbert Space Divergence

We can now utilize this newly established mixed conjugate framework to bridge a connection between the induced pseudo $f$–divergences and Bayes Hilbert spaces. One locally non-convex function we already considered in Table 1 is closely related to the centered log-ratio transformation and of special interest for the remainder of the paper. It is defined as

$$f_{BHS} : [0, \infty) \to \mathbb{R}, \quad x \mapsto x \log(x)^2. \tag{5}$$

It is easy to verify that $f_{BHS}$ is concave on the interval $[0, e^{-1}]$ as well as $f_{BHS}(1) = 0$. We can therefore use $f_{BHS}$ to define the pseudo $f$-divergence $D_{f_{BHS}}$.

Let $\mu \in B^2(\lambda)$, then we can derive a connection between this newly constructed divergence and the Bayes Hilbert space metric:

**Lemma 4.5.** *For $f_{BHS}$ as defined in equation 5, it holds that*

$$D_{f_{BHS}}(\mu, \nu) = d_{B^2(\mu)}(p_\mu, p_\nu)^2 + \mathbb{E}_\mu \left( \log(\mu \ominus \nu) \right)^2.$$

Minimizing $D_{f_{BHS}}$ also minimizes $d_{B^2(\mu)}(p_\mu, p_\nu)$, ensuring that two probability measures close with respect to $D_{f_{BHS}}$ are similarly close in the corresponding Bayes Hilbert space. This aligns intuitively with statistical learning, where we assume $\mu$ as the true underlying measure and aim to learn a measure $\nu$ that closely approximates it. Consequently, $\mu$ generates a Bayes Hilbert space $B^2(\mu)$ in which the approximations must reside. Since densities relative to $\mu$ are rarely of primary interest, we convert them to more familiar densities, such as those with respect to Lebesgue measures.

The mixed conjugate of $f_{BHS}$ is given by

$$f_{*,\text{BHS}}^*(x) = \begin{cases} 2(-1 - \sqrt{1+x}) \exp(-1 - \sqrt{1+x}) & x \in [-1, 0) \\ 2(-1 + \sqrt{1+x}) \exp(-1 + \sqrt{1+x}) & x \geq 0. \end{cases}$$

Here, $M = [0, \infty)$ and $N = [-1, 0)$. Applying Lemma 4.1 implies that $f^{**}_{*,\text{BHS}} = f_{\text{BHS}}$ and Theorem 4.3 implies that $\tilde{T}(x) = f'_{BHS}\left(\frac{p_\mu(x)}{p_\nu(x)}\right)$ is and optimizer for equation 3. Since $f^*_{*,\text{BHS}} \circ \tilde{T}(x) = 2\log(x)x$, it follows that

$$
\begin{aligned}
\sup_{T \in C(M^*)} &\left\{ (\mathbb{E}_\mu(T) - \mathbb{E}_\nu(f^*_* \circ T))\mathbb{1}_{\{T \in M_C\}} \right\} + \inf_{T \in C(N^*)} \left\{ (\mathbb{E}_\mu(T) - \mathbb{E}_\nu(f^*_* \circ T))\mathbb{1}_{\{T \in N_C\}} \right\} \\
&= \mathbb{E}_\mu(\tilde{T}) - \mathbb{E}_\nu(f^*_* \circ \tilde{T}) \\
&= D_{f_{\text{BHS}}}(\mu, \nu).
\end{aligned}
$$

Thus, the bound in Corollary 4.4 is tight for $f_{BHS}$. In summary, we have demonstrated that despite the local non-convexity of $f_{BHS}$, it defines a pseudo $f$-divergence closely related to the Bayes Hilbert space metric. A slight adaptation of the results from Nowozin et al. (2016); Nguyen et al. (2010) enables the minimization of a lower bound for $D_{f_{BHS}}$, with this bound being attained by the optimal estimate of $\tilde{T}$.

In the next section, we will apply these results to generative modeling, introducing a new pseudo $f$-GAN (Bayes Hilbert space GAN) that enables sampling from a learned Bayes Hilbert space distribution.

## 5 COMPUTATIONAL RESULTS

### 5.1 SETUP

Using the PyTorch framework (Paszke et al., 2019), we implemented the Bayes Hilbert space GAN (BHSGAN) and compared its results to traditional $f$-GANs, shown in Table 2, as well as a Wasserstein-GAN (Arjovsky et al., 2017) on the MNIST (Deng, 2012) (Appendix A.2) and CIFAR10 (Krizhevsky et al., 2009) data sets. Furthermore, we trained the benchmarking GAN architectures to approximate a heavy-tailed distribution, specifically the lognormal distribution with parameters $(\text{loc} = 0, \text{shape} = 0.5, \text{scale} = 1)$.

Table 2: Specification of $f$-GAN activation functions and Fenchel conjugates similar to Nowozin et al. (2016).

| Name | $g_f$ | dom$(f^*_{(*)})$ | (Mixed) Conjugate $f^*_{(*)}(t)$ |
|---|---|---|---|
| Kullback-Leibler (KL) | $v$ | $\mathbb{R}$ | $\exp(t-1)$ |
| Reverse KL | $-\exp(-v)$ | $\mathbb{R}_-$ | $-1 - \log(-t)$ |
| Pearson | $v$ | $\mathbb{R}$ | $\frac{1}{4}t^2 + t$ |
| GAN | $-\log(1+\exp(-v))$ | $\mathbb{R}_-$ | $-\log(1 - \exp(t))$ |
| BHS | $-1 + \exp(v)$ | $[-1, \infty)$ | $2\left(-1 \pm \sqrt{1+t}\right)\exp\left(-1 \pm \sqrt{1+t}\right)$ |

In terms of implementation, our approach closely resembles Nowozin et al. (2016) where the authors estimate generative models using variational divergence minimization (Nguyen et al., 2010). According to Corollary 4.4, we need to optimize equation 4 with respect to $\bar{T}$, which is equivalent to optimizing

$$
\min_\vartheta \max_\omega F(\vartheta, \omega) := \min_\vartheta \max_\omega \mathbb{E}_\mu(\bar{T}_\omega) - \mathbb{E}_{\nu_\vartheta}(f^*_* \circ \bar{T}_\omega). \tag{6}
$$

Here, $\nu_\vartheta$ represents the generative model taking a random vector as an input and returning a sample to be evaluated by the discriminatory model. The discriminatory model $\bar{T}_\omega$ returns some real-valued constant we call *score*. A high value for the score indicates that the discriminatory model is confident of the sample being generated by the generative model. In contrast, a low score indicates that the discriminatory model evaluated the underlying sample to stem from the training data. $\vartheta$ and $\omega$ denote the parametrizations of the respective models. We optimize equation 6 by sampling from a finite training data set to estimate the underlying distribution. As underlying models for the approximation on the MNIST and synthetic heavy tailed data set, we chose basic feed-forward neural networks identical to the ones used in Nowozin et al. (2016).

For the CIFAR-10 dataset, we employ the DCGAN architecture introduced by Radford et al. (2015). We also utilize the same representation of the variational functions as in Nowozin et al. (2016), summarized as follows: To ensure that $\bar{T}_\omega$ respects the domain of $f_*^*$, we represent the variational function as $\bar{T}_\omega$ by $\bar{T}_\omega(x) = g_f \circ V_\omega(x)$. Here, $V_\omega$ denotes the discriminatory model with no restrictions on the output, meaning $\mathrm{Im}(V_\omega) \subseteq \mathbb{R}$ while $g_f$ is a final output activation function tailored to the domain of $f^*$ (see Table 2). This approach accommodates various $f$-divergences, particularly our newly introduced approach that allows for a locally non-convex divergence-generating function. In our specific case, where $f(x) = x \log(x)^2$, we choose $g_f(x) = -1 + \exp(v)$, to ensure the the output lies in $[-1, \infty)$. To train the generative adversarial network, we slightly adjust the approach in Radford et al. (2015). The specific adjustments, configurations, and algorithm can be found in the appendix.

## 5.2 DISCUSSION OF RESULTS

To evaluate the performance of each GAN architecture in terms of estimating a heavy-tailed distribution, we estimate the shape and scale parameters of the generated distribution. These estimates are compared to the true parameters to assess each model's ability to capture the distribution's heavy-tailed nature.

Table 3 indicates that BHSGAN approximates the true shape parameter of the lognormal distribution more accurately than every other architecture. While the KL GAN slightly outperforms the BHS GAN in scale estimation, KL GAN consistently overestimates the shape parameter, with an average value nearly twice the true parameter. Thus, BHSGAN emerges as a robust choice for generating heavy-tailed distributions like the lognormal distribution.

Table 3: Comparison of GAN architectures with respect to shape and scale parameters.

| Name | shape = 0.5 | scale = 1 |
|---|---|---|
| BHSGAN | $0.5049 \pm 0.0104$ | $1.0937 \pm 0.0227$ |
| Wasserstein | $0.4220 \pm 0.0071$ | $1.6670 \pm 0.0127$ |
| GAN | $0.1952 \pm 0.0028$ | $1.3422 \pm 0.0130$ |
| Pearson | $0.6181 \pm 0.0240$ | $0.6863 \pm 0.0183$ |
| KL | $0.9951 \pm 0.0167$ | $1.0126 \pm 0.0207$ |
| Reverse KL | $0.8949 \pm 0.0178$ | $0.7376 \pm 0.0120$ |

In addition to the descriptive analysis, we provide and discuss a histogram and KDE estimate for each GAN architecture in Appendix A.2, which allows for a visual comparison of the different generated distributions.

However, the performance of the BHSGAN on the CIFAR10 dataset can be assessed by comparing the Fréchet Inception Distance (FID) (Heusel et al., 2017) of generated samples to different $f$-GAN and Wasserstein GAN architectures. Table 4 summarizes FID scores of the tested architectures.

The BHSGAN outperforms all $f$-GAN variants, proving its effectiveness in generating higher-quality and more diverse images. While the WGAN achieves a marginally better FID score, the difference is negligible and unlikely to be conclusive. Moreover, our approach guarantees that the approximated and true distributions remain closely aligned in the Bayes Hilbert space.

Upon inspecting Figure 2, the BHSGAN architecture stands out among the other $f$-GAN architectures by having the most refined picture structures. The KL, Reverse KL, and Pearson GAN produce less recognizable shapes using the same training procedure as the others.

Table 4: Mean FID Scores and two times the standard deviation for different architectures on the CIFAR10 data set over 100 seeds. Lower FID scores indicate better model performance.

| MODEL | FID |
|---|---|
| BHSGAN | $31.26 \pm 0.08$ |
| KL GAN | $37.50 \pm 0.13$ |
| REVERSE KL GAN | $85.27 \pm 0.19$ |
| PEARSON GAN | $33.60 \pm 0.38$ |
| GAN | $33.60 \pm 0.10$ |
| WGAN | $30.81 \pm 0.12$ |

As shown in Table 4, the WGAN and BHSGAN architectures produce the most recognizable samples. We can therefore conclude, that the BHSGAN outperforms traditional $f$-GANs and is on par with the Wasserstein architecture.

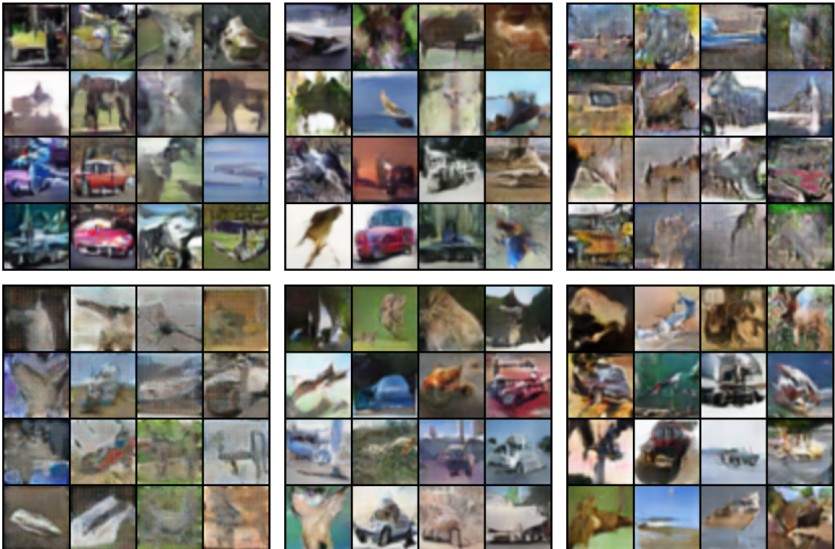

Figure 2: Experimental results for the CIFAR10 data set: Top row, from left to right: BHSGAN, KLGAN, and Reverse KLGAN. Bottom row, from left to right: Pearson GAN, original GAN, and Wasserstein GAN. Samples were drawn from the generators trained on the CIFAR10 dataset.

## 6 CONCLUSION

In this paper, we presented a novel framework for generalizing $f$-divergences, wherein we proved that the divergence-generating function of an $f$-divergence does not need to be convex on the whole domain, allowing for an even larger class of divergence-generating functions. We then showed, that there is a direct connection between the metric induced by the Bayes Hilbert space inner product and a pseudo $f$-divergence with pseudo-divergence-generating function $f_{BHS}$.

This connection of $f$-divergences and Bayes Hilbert spaces poses an interesting future direction as it circumvents one of the drawbacks of Bayes Hilbert space learning: learning from samples from distributions. Additionally, our framework enables sampling from estimated Bayes Hilbert space distributions in high dimensional settings, a feature that functional methods of estimation lack by design due to the curse of dimensionality. The framework therefore poses an interesting approach for sampling from posterior distributions in Bayesian statistics, which has to be addressed in detail in future research.

Since the primary goal of this publication is to extend the theoretical framework of $f$-divergences and bridge the gap to Bayes Hilbert spaces, we want to improve GAN performance in future work. Performance in terms of FID-Scores could be improved by considering a more advanced training procedure like progressively growing GAN (Karras et al., 2017) or state-of-the-art architectures like StyleGAN (Karras et al., 2019) and VQGAN (Esser et al., 2021). The proposed framework can also easily be implemented as an addition to existing $f$-GAN implementations as it only requires defining the new pseudo-divergence generating function $f_{BHS}$ and adjusting the output activation function $g_f$ specific to the domain of the generator.

## 7 REPRODUCIBILITY STATEMENT

We have taken several steps to ensure the reproducibility of our work. In Section 5, we provide detailed explanations of the framework and model architectures, with pseudocode for the key algorithm included in A.3. Proofs of technical results can be found in Section A.1.

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

## A  APPENDIX / SUPPLEMENTAL MATERIAL

### A.1  PROOFS OF THEORETICAL RESULTS

In this section, we provide proofs of several results we introduced throughout the publication.

While the following Lemma does not concern BHSGANs directly, we noticed that this connection has not yet been drawn in previous publications.

*Proof of Lemma 3.1.*

$$
\begin{aligned}
d_{B^2(\lambda)}(p_\mu, p_\nu)^2 &= \langle p_\mu \ominus p_\nu, p_\mu \ominus p_\nu \rangle_{B^2(\lambda)} \\
&= \langle \mathrm{clr}(p_\mu \ominus p_\nu), \mathrm{clr}(p_\mu \ominus p_\nu) \rangle_{L^2(\lambda)} \\
&= \int_\Omega \log(p_\mu(x) \ominus p_\nu(x))^2 \lambda(\mathrm{d}x) - \left( \int_\Omega \log(p_\mu(x) \ominus p_\nu(x)) \lambda(\mathrm{d}x) \right)^2 \\
&= \mathrm{Var}_\lambda(\log(\mu \ominus \nu)).
\end{aligned}
$$

$\square$

The proof of Lemma 4.1 is straightforward but shows nevertheless the importance of the restriction to functions $f$ that are unbounded on the convex part of the domain.

*Proof of Lemma 4.1.*

1.  (a) Let $t^* \in M$. Then $\hat{x} := \hat{x}(t^*) = \underset{x \in \mathrm{dom}(f)}{\mathrm{argmax}} (xt^* - f(x)) \in [a, \infty)$. Since $\hat{x} \in [a, \infty)$, the function $x \mapsto xt^* - f(x)$ is concave in $\hat{x}$ and monotonically decreasing for any $x > \hat{x}$. Thus, $\underset{x \to \infty}{\lim} xt^* - f(x) = -\infty$, i.e., $\underset{x \in \mathrm{dom}(f)}{\mathrm{argmin}} \, xt^* - f(x) \in [a, \infty)$ implying that $t^* \notin N$. Analogously, $t_* \in N \implies t_* \notin M$.
    (b) In order to show that $M \cup N = \mathrm{dom}(f_*^*)$, let $\tilde{t} \in \mathrm{dom}(f_*^*)$. Then, there exists a $x_{\tilde{t}}$ such that $f_*^*(\tilde{t}) = x_{\tilde{t}}\tilde{t} - f(x_{\tilde{t}})$. Since $x_{\tilde{t}} \in \mathrm{dom}_f$, either $x_{\tilde{t}} \in [0, a)$ or $[a, \infty)$. As $\tilde{t}$ was chosen arbitrarily, $M \cup N = \mathrm{dom}(f_*^*)$ holds.

2.  Since $M$ and $N$ are disjoint, we can write

$$
\begin{aligned}
f_{**}^{**} &= \underset{x \in M}{\sup} \{tx - f_*^*(x)\} \mathbb{1}_{\{t \in M^*\}} + \underset{x \in N}{\inf} \{tx - f_*^*(x)\} \mathbb{1}_{\{t \in N^*\}} \\
&= \underset{x \in M}{\sup} \{tx - f^*(x)\} \mathbb{1}_{\{t \in M^*\}} + \underset{x \in N}{\inf} \{tx - f_*(x)\} \mathbb{1}_{\{t \in N^*\}} \\
&= f\mathbb{1}_{\{t \in M^*\}} + f\mathbb{1}_{\{t \in N^*\}} \\
&= f.
\end{aligned}
$$

$\square$

In Lemma 4.2 we restrict ourselves to the class of functions that have a nonnegative domain and range. The reason for this restriction is simply to ensure that the introduced measure of dissimilarity respects homogeneity.

*Proof of Lemma 4.2.* Let $\mu, \nu \in B^2(\lambda)$. Then,

$$
\begin{aligned}
D_f(\mu, \nu) &= \int_\Omega p_\nu(x) f\left( \frac{p_\mu(x)}{p_\nu(x)} \right) \lambda(\mathrm{d}x) \\
&= \int_\Omega \underbrace{p_\mu(x)}_{\geq 0} \underbrace{f\left( \frac{p_\mu(x)}{p_\nu(x)} \right)}_{\geq 0} \lambda(\mathrm{d}x) \\
&\geq 0
\end{aligned}
$$

and

$$D_f(\mu,\nu) = \int_\Omega f\left(\frac{p_\mu(x)}{p_\nu(x)}\right)\nu(\mathrm{d}x) \overset{!}{=} 0 \iff \mu = \nu \quad \lambda\text{-a.s.}$$

$\square$

In order to prove Theorem 4.3, we have to resort to a more technical argument involving the fundamental lemma of the calculus of variations to ensure that we can indeed derive an analytical representation for the optimizer of equation 3.

*Proof of Theorem 4.3.* In order to optimize equation 3, we have to maximize

$$\left(\mathbb{E}_\mu(T) - \mathbb{E}_\nu(f_*^* \circ T)\right)\mathbb{1}_{\{T \in C(M^*)\}}$$

and minimize

$$\left(\mathbb{E}_\mu(T) - \mathbb{E}_\nu(f_*^* \circ T)\right)\mathbb{1}_{\{T \in C(N^*)\}}$$

with respect to some $T \in C(\mathrm{dom}(f_*^*))$. Consider the term

$$\mathbb{E}_\mu(T) - \mathbb{E}_\nu(f_*^* \circ T).$$

It can be rewritten as

$$\mathbb{E}_\mu(T) - \mathbb{E}_\nu(f_*^* \circ T) = \int_0^\infty T(x)\mu(\mathrm{d}x) - \int_0^\infty (f_*^* \circ T)(x)\nu(\mathrm{d}x)$$

$$= \underbrace{\int_0^\infty T(x)p_\mu(x) - (f_*^* \circ T)(x)p_\nu(x)\lambda(\mathrm{d}x)}_{:=E(T)}.$$

Then, for an optimizer of equation 3

$$\left.\frac{\partial}{\partial\varepsilon}\right|_{\varepsilon=0} E(T+\varepsilon U) \overset{!}{=} 0$$

with $U \in C_0^\infty$ is a necessary condition for an extremum (Gelfand et al., 2000, Theorem 4.1). Now,

$$E(T+\varepsilon U) = \int_0^\infty \left[(T+\varepsilon U)(x)p_\mu(x) - f_*^* \circ (T+\varepsilon U)(x)p_\nu(x)\right]\lambda(\mathrm{d}x)$$

and

$$\frac{\partial}{\partial\varepsilon}E(T+\varepsilon U) = \frac{\partial}{\partial\varepsilon}\int_0^\infty \left[(T+\varepsilon U)(x)p_\mu(x) - f_*^* \circ (T+\varepsilon U)(x)p_\nu(x)\right]\lambda(\mathrm{d}x)$$

$$= \int_0^\infty \left[U(x)p_\mu(x) - (f_*^*)'(T(x)+\varepsilon U(x))U(x)p_\nu(x)\right]\lambda(\mathrm{d}x)$$

$$= \int_0^\infty U(x)\left[p_\mu(x) - (f_*^*)'(T(x)+\varepsilon U(x))p_\nu(x)\right]\lambda(\mathrm{d}x)$$

$$\overset{\varepsilon=0}{=} \int_0^\infty U(x)\left[p_\mu(x) - (f_*^*)'(T(x))p_\nu(x)\right]\lambda(\mathrm{d}x). \tag{7}$$

Applying the fundamental lemma of the calculus of variations (Gelfand et al., 2000, Lemma 1.1) to equation 7 yields

$$\int_0^\infty U(x)\left[p_\mu(x) - (f_*^*)'(T(x))p_\nu(x)\right]\lambda(\mathrm{d}x) = 0 \implies p_\mu(x) - (f_*^*)'(T(x))p_\nu(x) = 0 \quad \text{a.e.}.$$

Thus,

$$T(x) = \left((f_*^{*\prime})^{-1} \circ \frac{p_\mu}{p_\nu}\right)(x)$$

is a candidate for an optimizer. Note, that if $(f_*^{*\prime})^{-1}$ is well-defined, then $(f_*^{*\prime})^{-1}(x) = f'(x)$ (cf. Rockafellar (1970)). For the second derivative, we have

$$\frac{\partial^2}{\partial\varepsilon^2}\bigg|_{\varepsilon=0} E(T+\varepsilon U) = \int_0^\infty -(f_*^*)''(T(x))\underbrace{U(x)^2 p_\nu(x)}_{\geq 0}\lambda(\mathrm{d}x).$$

Depending on the properties of $f_*^*$ the candidate

$$\tilde{T}(x) = \left((f_*^{*\prime})^{-1} \circ \frac{p_\mu}{p_\nu}\right)(x)$$

is thus a maximizer or minimizer. The restriction in the maximization part of equation 3 enforces that the optimizer $\tilde{T}$ is convex, implying that

$$(f_*^*)'' \circ \left((f_*^{*\prime})^{-1} \circ \frac{p_\mu}{p_\nu}\right)(x) \geq 0$$

and therefore

$$\frac{\partial^2}{\partial\varepsilon^2}\bigg|_{\varepsilon=0} E(T+\varepsilon U) \leq 0.$$

Conversely, to optimize the minimization part of equation 3, we consider the restriction to the concave part of $f_*^*$, implying that

$$(f_*^*)'' \circ \left((f_*^{*\prime})^{-1} \circ \frac{p_\mu}{p_\nu}\right)(x) \leq 0$$

and therefore

$$\frac{\partial^2}{\partial\varepsilon^2}\bigg|_{\varepsilon=0} E(T+\varepsilon U) \geq 0.$$

$\square$

Corollary 4.4 plays an important role when considering the computational perspective of equation 3. Since we wanted to avoid optimizing a *min max* and *min min* game simultaneously, we derived a lower bound for the pseudo $f$ divergence that can be solved as a simple *min max* game instead. The core idea is to rewrite the infimum as a supremum and combine both suprema to obtain a *min max* game framework.

*Proof of Corollary 4.4.*

$$D_f(\mu,\nu) = \int_\Omega p_\nu(x) f\left(\frac{p_\mu(x)}{p_\nu(x)}\right)\lambda(\mathrm{d}x)$$

$$= \int_\Omega \sup_{t\in M}\left\{t\frac{p_\mu(x)}{p_\nu(x)} - f_*^*(t)\right\}\lambda(\mathrm{d}x) + \int_\Omega \inf_{t\in N}\left\{t\frac{p_\mu(x)}{p_\nu(x)} - f_*^*(t)\right\}\lambda(\mathrm{d}x) \quad (8)$$

$$\geq \sup_{T\in C(M^*)}\left\{\mathbb{E}_\mu(T) - \mathbb{E}_\nu(f_*^* \circ T)\right\} + \int_\Omega \inf_{T\in C(N^*)}\left\{t\frac{p_\mu(x)}{p_\nu(x)} - f^*(t)\right\}\lambda(\mathrm{d}x) \quad (9)$$

$$= \sup_{T\in C(M^*)}\left\{(\mathbb{E}_\mu(T) - \mathbb{E}_\nu(f_*^* \circ T))\right\} - \sup_{T\in C(N^*)}\left\{(\mathbb{E}_\mu(T) - \mathbb{E}_\nu(f_*^* \circ T))\right\} \quad (10)$$

$$= \sup_{T\in C(\mathrm{dom}(f_*^*))}\left\{\mathbb{E}_\mu(T(\mathbb{1}_{\{T\in C(M^*)\}} - \mathbb{1}_{\{T\in C(N^*)\}}))-\right.$$

$$\left.\mathbb{E}_\nu((f_*^* \circ T)(\mathbb{1}_{\{T\in C(M^*)\}} - \mathbb{1}_{\{T\in C(N^*)\}}))\right\}$$

$$= \sup_{\bar{T}\in C(M^*)\cup C(N^*)}\left\{\mathbb{E}_\mu(\bar{T}) - \mathbb{E}_\nu(f_*^* \circ \bar{T})\right\} \quad (11)$$

where $C(M^*)$ $(C(N^*))$ denotes the set of continuous functions such that $f_*^* \circ T$ is convex (concave). The second equality, equation 8, can be derived by applying the definition of the mixed conjugate.

equation 9 is derived in the same way as equation 1 and equation 10 is an application of Gossez et al. (2006) where we used that $-\inf = \sup$. In equation 11 we used the fact, that $C(M^*) \cap C(N^*) = \emptyset$. Defining $\bar{T} = T(\mathbb{1}_{\{T \in C(M^*)\}} - \mathbb{1}_{\{T \in C(N^*)\}})$ yields the desired result.

$\square$

*Proof of Lemma 4.5.*

$$
\begin{aligned}
D_{f_{BHS}}(\mu, \nu) &= \int_\Omega p_\nu(x) \frac{p_\mu(x)}{p_\nu(x)} \log\left(\frac{p_\mu(x)}{p_\nu(x)}\right)^2 \lambda(\mathrm{d}x) \\
&= \int_\Omega p_\mu(x) \log\left(\frac{p_\mu(x)}{p_\nu(x)}\right)^2 \lambda(\mathrm{d}x) \\
&= \mathbb{E}_\mu\left(\log(\mu \ominus \nu)^2\right) \\
&= d_{B^2(\mu)}(p_\mu, p_\nu)^2 + \mathbb{E}_\mu\left(\log(\mu \ominus \nu)\right)^2.
\end{aligned}
$$

$\square$

## A.2 Additional Experiments

In this section we present generated MNIST digits as well as histograms and KDEs for an estimated lognormal distribution for each benchmarking GAN architecture.

The findings of Table 3 can be validated further by considering Figure 3. BHS GAN approximates the true density well regarding mode and tail behavior. In comparison, the Wasserstein architecture fails to capture the left tail behavior completely, and the right tail shows some discrepancies in terms of monotonicity. While the KL GAN captures the true distributions' scale relatively well, the right tail seems to be much heavier than the tail of the true underlying distribution. Pearson GAN and Reverse KL GAN struggle with oscillations and deviations from the true density, especially in the tails. The basic GAN performs poorly in terms of alignment, with significant underestimation near the mode and noisy approximation in the tails.

The generated MNIST samples in Figure 4 visibly depict digits between $0$ and $9$, indicating that all GAN architectures learned the underlying distribution of the images well. Therefore we can conclude that the BHSGAN is on par with the other GAN architecture for the MNIST dataset. We limit ourselves to this high-level evaluation as a quantitative comparison for generative models on this kind of data is not meaningful (Nowozin et al., 2016).

## A.3 Practical considerations and algorithms

During the experiments we found that training (pseudo) $f$-GANs is highly unstable, suffering from exploding and diminishing losses. To stabilize the training process we implemented two methods. Firstly, we introduced a gradient penalty identical to the one used for WGANs (Gulrajani et al., 2017) for all used GANs except the vanilla and reverse KL-GAN. Additionally, contrary to the popular approach of training the discriminatory model first, we decided to start by training the generative model first. We interpret this finding as a (prior) practical drawback of $f$-GANs, in the way that in theory, the supremum has to be found over all continuous functions, but in practice, this class of functions is too general to be learned during training. This means that if the discriminator is allowed to be too flexible it starts "remembering" real images before the generator can generate good fake ones, resulting in exploding losses for the discriminator and vanishing ones for the generator. Accordingly, forcing the discriminator to be Lipschitz-continuous constrains the supremum and infimum in 3 to be found over all Lipschitz-continuous functions, which stabilizes the training process by hindering the discriminator from remembering real images too fast.

All experiments were conducted on an AORUS RTX 4090 eGPU 24GB.

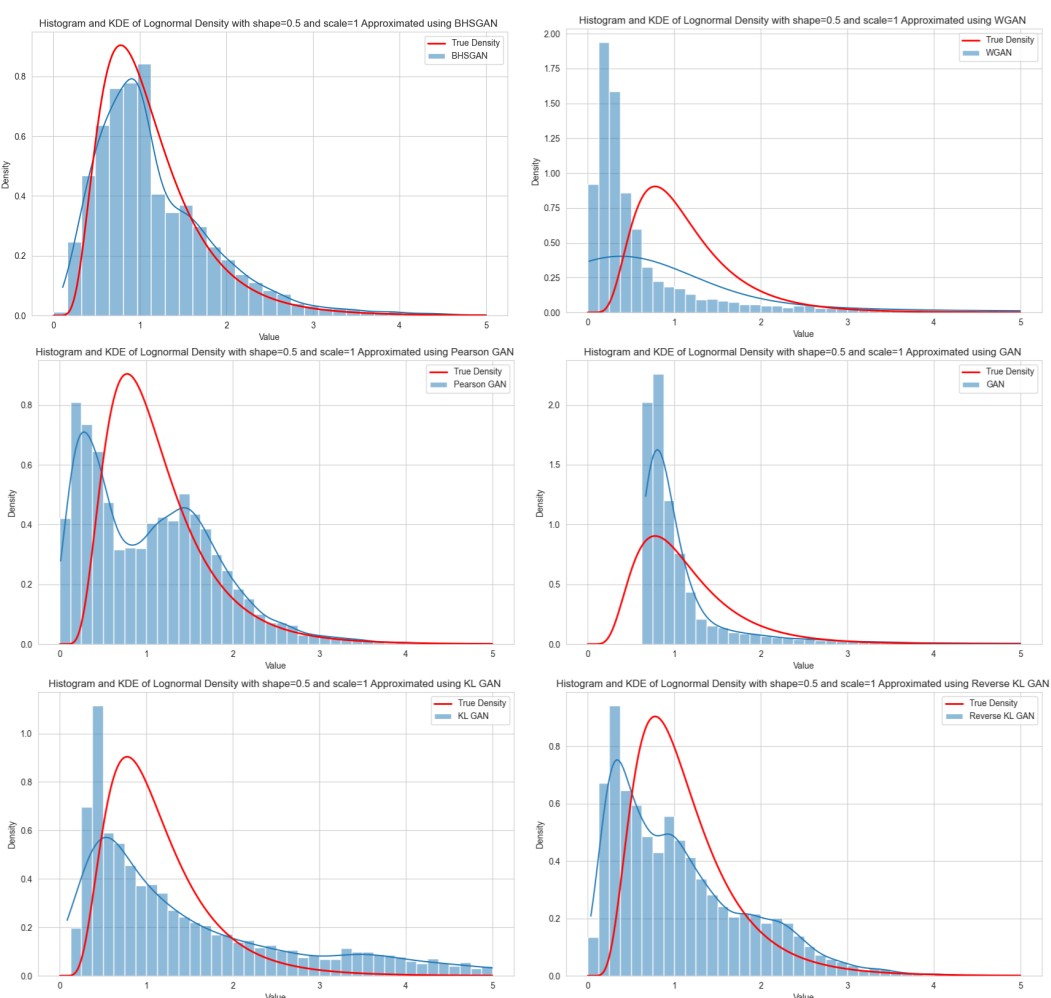

Figure 3: Histograms and KDE estimates for different GAN architectures

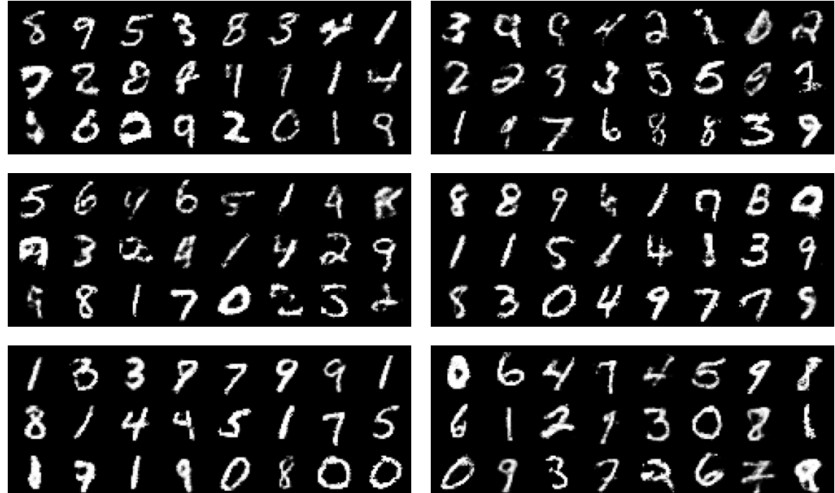

Figure 4: Experimental results for MNIST datasets: Left column, from top to bottom: BHSGAN, KLGAN, and Reverse KLGAN. Right column, from top to bottom: Pearson GAN, original GAN, and Wasserstein GAN. Samples were drawn from the generators trained on the MNIST dataset.

---

**Algorithm 1** Default values for the training procedure in this paper were $\alpha_g = \alpha_d = 0.0002$ for all GANs except BHSGAN and REVKLGAN, where $\alpha_g = \alpha_d = 0.00005$, $n_d = 2$, $n_g = 2$, $m = 128$ for MNIST and $m = 64$ for CIFAR, $N_{\text{epochs}} = 50$, $d_z = 100$ and $\lambda = 10$ except for PEARSONGAN, where $\lambda = 20$

---

**Input:** $\alpha_g$, learning rate of generative model. $\alpha_d$, learning rate of discriminatory model. $n_d$, number of discriminator updates. $n_g$, number of generator updates. $m$, batch size. $N_{\text{epochs}}$, number of epochs. $d_z$, dimension of noise, $\lambda$, gradient penalty coefficient, $gp$, gradient penalty, $w_0$, initial parameters of discriminatory model $\vartheta_0$, initial parameters of generative model.

**for** $k = 1, ..., N_{\text{epochs}}$ **do**
    **for each** batch $= \left\{ x^{(1)}, ..., x^{(m)} \right\}$ in training set **do**
        noise $= \left\{ z^{(1)}, ..., z^{(m)} \right\} \overset{i.i.d.}{\sim} \mathcal{N}(0, I_{d_z \times d_z})$
        real_sample $= \left\{ x^{(1)}, ..., x^{(m)} \right\}$
        **for** $n = 1, ..., n_g$ **do**
            $\mathcal{L}_{\nu_\vartheta} = -\frac{1}{m} \sum_{i=1}^{m} f_*^* \circ g_f \circ V_\omega \circ \nu_\vartheta(z^{(i)})$
            $\vartheta = \vartheta - \text{Adam}(\alpha_g, \vartheta, \mathcal{L}_{\nu_\vartheta})$
        **end for**
        **for** $n = 1, ..., n_d$ **do**
            $\mathcal{L}_{V_\omega} = \frac{1}{m} \sum_{i=1}^{m} \big( f_*^* \circ g_f \circ V_\omega \circ \nu_\vartheta(z^{(i)}) -$
            $\qquad\qquad\qquad g_f \circ V_\omega(x^{(i)}) \big) + \lambda \cdot gp$
            $\omega = \omega + \text{Adam}(\alpha_d, \omega, \mathcal{L}_{V_\omega})$
        **end for**
    **end for**
**end for**

---

