# OpenReview forum: "Bridging the Gap Between f-divergences and Bayes Hilbert Spaces"
_ICLR.cc/2025/Conference — ICLR 2025 Poster_

### Official Review · Reviewer_Y69u · 2024-10-20

**Soundness:** 3
**Presentation:** 3
**Contribution:** 2
**Rating:** 6
**Confidence:** 4

**Summary:**

This paper introduces a new loss for GANs, which builds on a connection between so-called Bayes Hilbert spaces and pseudo f divergences. First, Bayes Hilbert space are introduced, which is a linear space with additive and multiplicative structure coherent with Bayes theorem. Then, pseudo f divergences, which do not require f to be convex everywhere are introduced. The main part is that for a certain choice of f. the pseudo f divergence can be written as a sum of the BHS norm and some other 'entropy' term. Then, it is verified numerically, how well this certain choice of GAN performs in image generation.

**Strengths:**

The paper deals with a niche topic, but does it so very nicely. The definitions and theorems are perfectly clear, although I did not check the proofs. The connection between pseudo f divergences and Bayes Hilbert spaces is very interesting. The paper comes with code and seems reproducible. It is also nice that the paper rediscovers the log variance loss used in Richter et al and motivates it more theoretically.

**Weaknesses:**

1) My biggest problem apart from some nitpicks is that I do not really see the point of all this formalism. It is nice that there is an intrinsic connection between pseudo f divergences and Bayes Hilbert spaces, but there are no additional insights based on that. One could just introduce pseudo f divergences and use the specific f. One thing that seems a bit odd and might yield additional insights is that in Lemma 4.5, there is an additional entropy like term. Does this give any other interpretation?

2) Numerically, the results are okayish, mostly due to the use of the GAN algorithm. Unfortunately GANs suffer from many problems: unstable training, mode collapse,... Did you find if your loss alleviates some of these problems?

3) The interpretation of the numerical results is also a bit bold: The difference between WGAN and BHS-GAN "is on a par", while the BHSGAN "outperforms traditional f-GANs". Looking at the table, f-GANs (Pearson) are 2 FID away from BHS-GAN, so I would say there is not really anything conclusive about outperforming. In particular, the table states that the results are "over 100 seeds", which i interpret as you draw with one trained model a lot of images and calculate the FID and average this 100 times? Then, to say anything conclusive, one should try to get different training runs. However, I see this paper more of a theory paper and therefore I would suggest to tone the claims a bit down.

**Questions:**

1) Can the connection between Bayes Hilbert spaces and f divergences be leveraged for other generative models? (GANs are not in fashion anymore, due to well-known stability, mode collapse and interpretability reasons). Many generative models have a loss that can be bounded via the KL. I guess this goes in the direction of Richter et al.?

2) Bayes Hilbert spaces are introduced as they adhere to the structure of Bayes theorem, i.e., the likelihood is the radon nikodym derivative of posterior with prior. Can you somehow use these ideas to reformulate the distance between posterior and generation measure in terms of BHS losses?

---

> ### Author Response · Authors · 2024-11-19
> **Response to Weaknesses and Questions 1/3**
>
> Dear reviewer, first of all, thank you for reading our manuscript carefully and highlighting our potential shortcomings. A reply to your question and concerns can be found below.
>
> ## Weaknesses
>
> > [...] but there are no additional insights based on that.
>
> To establish this connection, we rigorously extended the existing $f$-divergence framework, which can also be viewed as our most important contribution.
> However, the initial motivation did not stem from explicitly extending this framework.
> We initially discovered the similarity between the Bayes Hilbert space (BHS) metric and $f$-divergences but quickly learned that the BHS metric is not part of this family due to the local non-convexity.
> The focus of our contribution, therefore, shifted more to developing a tool that allows us to draw this connection instead of simply stating it.
>  In our opinion, the proposed articles pave the way for further research with respect to this connection, both in theory and application, which we want to highlight with the following example:
>  In the works of Wynne [2023], the authors measure discrepancies for measures in BHS using KL Bayesian coresets.
> The use of our newly developed BHS divergence would allow a straightforward discrepancy measure in the same underlying BHS.
> The advantage of this discrepancy measure is that measures that are close with respect to $D_{f_{BHS}}$ are close in the corresponding BHS.
> Further applications and improvements of our newly developed theory are highlighted in the last paragraph of our conclusion.
>
> > One could just introduce pseudo f divergences and use the specific f.
>
> We think that, in a nutshell, this is what our contribution could be reduced to.
> However, as pointed out in the answer to the previous concern, our motivation stems from connecting $f$-divergences with BHS, as these fields have not been directly connected in previous works.
> Further work can build on the proposed results to gain an even deeper understanding either of the BHS divergence directly or of the general class of pseudo $f$-divergences.
>
> > [...] there is an additional entropy like term. Does this give any other interpretation?
>
> Arguing with Lemma 4.2., we know that $D_{f_{BHS}}(\mu,\nu)=0$ if and only if $\mu = \nu$, $\lambda$-a.s.
> We can, therefore, conclude that if the measures coincide, not only the BHS metric vanishes but also the respective entropy-like term.
> Additionally, the term $\log(x)^2$ captures how the values of the likelihood ratio deviate not just in magnitude but in a logarithmic sense, which is particularly meaningful in compositional or ratio-based spaces.
> By squaring the logarithm, the term highlights the magnitude of deviations symmetrically (penalizing both increases and decreases equally).
> We can, therefore, view this term as an additional penalty term that complements the purely geometric component of the BHS metric.
>
> > Did you find if your loss alleviates some of these problems?
>
> By introducing a gradient penalty, we observed more stable training, including less mode collapse. However, this was applied to all GAN architectures and not only to the BHSGAN.
> We acknowledge that the GAN algorithm is relatively outdated, but as pointed out in the conclusion, the newly developed pseudo divergence can also be applied in more advanced architectures such as StyleGAN and VQGAN.
> Since our computational resources are rather limited we had to validate our finding by using simpler architectures.
>
> > The interpretation of the numerical results is also a bit bold [...]
>
> We thank the reviewer for their thoughtful feedback regarding the interpretation of our numerical results.
> We agree that our claims regarding the performance differences between BHS-GAN and traditional $f$-GANs, as well as WGAN, were presented too boldly and could have been more nuanced.
>
> The reviewer is correct in interpreting our statement about results over 100 seeds: the FID scores were indeed averaged across multiple draws with a single trained model rather than from multiple independent training runs.
> Our proposed framework leverages the fact that every trained model is fine-tuned to achieve the best possible FID score without suffering from mode collapse.
> We, therefore, feel that our evaluation is still conclusive in terms of model performance.
>
> As the primary contribution of our work lies in the theoretical framework, we propose changing the manuscript to appropriately tone down these performance claims. Specifically, we propose:
>
>  - Reframing the comparative statements to emphasize the general compatibility of BHS-GAN with other methods rather than asserting clear superiority.
> - Clearly outline the limitations of the current experimental setup, including the averaging procedure over a single model and the absence of results from multiple training runs.
> - Highlight the theoretical contributions of the work as the main focus while presenting the numerical results as supportive and exploratory.

---

> ### Author Response · Authors · 2024-11-19
> **Response to Weaknesses and Questions 2/3**
>
> Additionally, other reviewers pointed out that we should provide further experiments to demonstrate the performance of our model in terms of approximating heavy-tailed distributions. We therefore conducted an additional experiment which we added to the manuscript. A summary can be found below.
>
> ### Additional Experiments
> We train the benchmarking GAN architectures to approximate a heavy-tailed distribution, specifically the lognormal distribution with parameters $(\mathrm{loc}=0, \mathrm{shape}=0.5, \mathrm{scale}=1)$.
> The performance of each GAN is assessed by estimating the three parameters and using these estimates to analyze each model's ability to capture the heavy-tailed nature of the distribution.
>
> From a statistical point of view, the KS test is unsuitable in this context as it does not adequately account for tail behavior.
> Therefore, particular attention will be given to comparing the estimated parameter $\mathrm{shape}$ and $\mathrm{scale}$ to evaluate the frameworks' performance in modeling the distribution's tail behavior.
>
> The steps for this experimental setup can be summarized as follows:
>
> 1. We first generate 10.000 lognormally distributed samples with parameters $(\mathrm{loc}=0, \mathrm{shape}=0.5, \mathrm{scale}=1)$.
> 2. Then, we train each GAN architecture for $35$ epochs using the same set of hyperparameters.
> 3. To evaluate the training results, we first generated $1000$ realizations of the estimated distribution for each GAN architecture.
>     Then, we estimated the shape and scale parameters for each of the $1000$ realizations and compared standard descriptive statistics such as mean and standard deviation to the true parameters.
>
> The following table contains the result of this experiment:
>
> | Name          | shape = 0.5           | scale = 1           |
> |---------------|-----------------------|---------------------|
> | BHS           | 0.5049 ± 0.0104      | 1.0937 ± 0.0227     |
> | Wasserstein   | 0.4220 ± 0.0071      | 1.6670 ± 0.0127     |
> | GAN           | 0.1952 ± 0.0028      | 1.3422 ± 0.0130     |
> | Pearson       | 0.6181 ± 0.0240      | 0.6863 ± 0.0183     |
> | KL            | 0.9951 ± 0.0167      | 1.0126 ± 0.0207     |
> | Reverse KL    | 0.8949 ± 0.0178      | 0.7376 ± 0.0120     |
>
> The BHSGAN approximates the true shape parameter more accurately than every other architecture.
> While the KL GAN slightly outperforms the BHSGAN in scale estimation, KL GAN consistently overestimates the shape parameter, with an average value nearly twice the true parameter.
> Thus, the BHSGAN emerges as a robust choice for generating heavy-tailed distributions like the lognormal distribution.
>
> The findings of  the table above can be validated further by considering the newly added figure in the appendix.
> BHSGAN approximates the true density well regarding mode and tail behavior.
> In comparison, the Wasserstein architecture fails to capture the left tail behavior completely, and the right tail shows some discrepancies in terms of monotonicity.
> While the KL GAN captures the true distributions' scale relatively well, the right tail seems to be much heavier than the tail of the true underlying distribution.
> Pearson GAN and Reverse KL GAN struggle with oscillations and deviations from the true density, especially in the tails.
> The basic GAN performs poorly in terms of alignment, with significant underestimation near the mode and noisy approximation in the tails.
>
> ## Questions
>
> > Can the connection between Bayes Hilbert spaces and f divergences be leveraged for other generative models?
>
> We want to bring attention to the last paragraph of our proposed manuscript, where we discuss limitations and further research ideas.
>  In particular, we address that performance in terms of FID-Scores could be improved by considering a more advanced training procedure with state-of-the-art architectures like StyleGAN and VQGAN.
>
> > Many generative models have a loss that can be bounded via the KL.
>
> In the works of Richter et al. [2020], the authors focus on the evidence lower
> bound (ELBO) for variational inference, which involves the KL divergence.
> We suppose that depending on the underlying (pseudo) $f$-divergence, different lower bounds can be derived.
> In Corollary 4.4. we derive a lower bound for pseudo divergences that relies on a variational representation.
> In Wynne [2023], the authors also propose the KL divergence as a lower bound for the BHS norm: $KL(\mu,\nu)\leq 2\exp(B)\|\mu-\nu\|_{B^2(\mu)}$, where $B>0$ is some constant.
> The proposed bound could potentially be incorporated in our theory.
> However, we believe that this would lead to a significant increase in theory and should therefore be explored in a different publication.

---

> > ### Author Response · Authors · 2024-11-19
> > **Response to Weaknesses and Questions 3/3**
> >
> > > [...] Can you somehow use these ideas to reformulate the distance between posterior and generation measure in terms of BHS losses?
> >
> > Could the reviewer please elaborate on which connection to our proposed manuscript he has in mind? Based on our interpretation, we want to bring forward the following solution:
> > In a Bayes Hilbert space (BHS), the generating measure is the neutral element of addition and subtraction. Therefore, the BHS distance between this generating measure and the posterior is simply the BHS norm of the posterior. Speaking in terms of $D_{f_{BHS}}$ this results in the following formula where $f_{p}$ and $F_{p}$ denote the posterior density and measure respectively and $\mathbb{E}\lambda$ the expected value with respect to $\lambda$:
> >
> > $ D_{f_{BHS}}(\lambda,F_{p}) = \|f_{p}\|_{B^2(\lambda)}^2  $ $+\mathbb{E}\lambda (\log(f_p))^2$
> >
> > ## Final Remarks
> >
> > Again, we thank the reviewer for reading our manuscript carefully and suggesting improvements. We hope that our changes to the manuscript address the reviewer's concerns adequately.

---

> > > ### Comment · Reviewer_Y69u · 2024-11-19
> > > **Thanks!**
> > >
> > > Thank you for your response. With the last point I meant whether the BHS formulation can be used for conditional generation, as BHS structure is basically built for Bayes theorem. However, I am also not sure how this can be done, just a curiosity.
> > >
> > > In light of the response, I am raising my score to 6. Please clarify the confusing points in the manuscript.

---

### Official Review · Reviewer_Tns5 · 2024-10-28

**Soundness:** 3
**Presentation:** 3
**Contribution:** 2
**Rating:** 6
**Confidence:** 3

**Summary:**

This work proposes novel connections between Bayes Hilbert spaces and a modified version of $f$-divergences (namely pseudo $f$-divergences) requiring $f$ to be concave on $[0,a)$ and convex on $[a,+\infty), a>0$ instead of convexity on all $[0,+\infty)$. More precisely.

- Basic results and definition for both Bayes Hilbert spaces and $f$-divergences are recalled in Sections 2 and 3.
- Extensions of classical results of $f$-divergences for pseudo $f$-divergences are proposed in Section 4.1
- Authors unveil a connection between a specific pseudo $f$-divergence and a Bayes Hilbert Space
- Experiments on GANs are proposed in Section 5, showing the pseudo $f$-divergence approach on a certain learning problem outperforms the classical $f$-GAN approach and matches the performance of Wasserstein GANs

**Strengths:**

- The three first sections are well-written and pedagogical.
- The theoretical connection of Section 4.2 looks strong and promising.

**Weaknesses:**

- The precise nature of the contribution would gain to be clearer
- The experimental section is poorly written and does not exploit the flexibility provided by the proposed theoretical framework.

Please see the Questions for more details.

**Questions:**

- Can you precise why the addition, multiplication and substraction in the Bayes Hilbert is well-defined only if we consider $B(\lambda)$ and not $\mathcal{M}(\lambda)$ ?
- You said in the abstract that your framework 'generalizes f -divergences by incorporating locally non-convex divergence-generating functions'. This should be clarified as, to my understanding of Section 4, you only consider a certain type of local non-convexity, being functions concave on $[0,a)$ and convex on $[a,+\infty)$? Do you think it is possible to consider functions $f$ with several areas of non-convexity, eg  a function convex on $[0,a], [b,c], [d,+\infty)$ and concave on $(a,b), (c,d)$?
- Experiments: can you please detail the nature of your experiment? It is not clear to understand what you are doing and why you are doing it?
- You mentioned in your intro (l.104) that in case of heavy-tail distribution, Wasserstein distance is unsuitable to modeling distributions with heavy-tails. Given that your experiments shows similar performances with WGAN, it seems that you do not consider heavy-tailed distributions, could you think of an experiment (even a toy one) with such a heavy-tailed measure where the WGAN would fail and your approach succeed?
- l 431 typo for 'equation'
- I find the nature of your theoretical contribution unclear: you claim that you are bridging the gap between Bayes Hilbert space and $f$-divergences. However, you are not creating such a link as you focus only on pseudo divergences, with $f$ concave on $[0,a], a>0$ and convex on $[a,+\infty]$ with $a>0$. As this does not encompass the case $a=0$ (ie the case of a true $f$-divergence, if we add a lower semicontinuous assumption), I don't know whether:
	1. the case $a=0$ has already been dealt with in the literature, then it would be good to precise such links more clearly and that your contribution is an extension of those links to the case of pseudo-divergences
	2. The case $a=0$ has not been previously considered, but you were only able to design theoretical links for pseudo $f$-divergences. In this case you are not truly reaching $f$-divergences
	In both cases, claiming that you are bridging the gap between Bayes Hilbert space and $f$-divergences seems inaccurate.

---

> ### Author Response · Authors · 2024-11-19
> **Response to Weaknesses and Questions 1/3**
>
> Dear reviewer, first of all, thank you for reading our manuscript carefully and highlighting our potential shortcomings.
> A reply to your question and concerns can be found below.
>
> ## Weaknesses
>
> > The precise nature of the contribution would gain to be clearer.
>
> To address the reviewers' concern about the precise nature of our contribution, we would like to elaborate on our main contributions, which are summarized in the first section of our manuscript:
>
> To bridge the gap between Bayes Hilbert spaces (BHS) and $f$-divergences, we generalized $f$-divergence theory to accommodate local non-convexities in divergence-generating functions.
> The pseudo-divergence generating function $x\mapsto x\log(x)^2$, which exhibits such a non-convexity, directly relates to the BHS metric, as shown in Lemma 4.5.
> Besides proving the well-definedness of this newly defined pseudo divergence in Lemma 4.2., we show in Theorem 4.3. that the variational optimization problem for minimizing the pseudo divergence between two distributions has an optimizer that can be represented analytically.
> Additionally, we introduce a novel way of learning BHS distributions from samples, which has been challenging in the past (see Maier et al. [2021], Birrell et al. [2022]) and provide a mechanism of sampling from these distributions, which was not feasible before.
> In the experimental section, we compare FID scores to show that this newly developed framework is competitive with traditional architectures on image domains.
>
> > The experimental section is poorly written and does not exploit the flexibility provided by the proposed theoretical framework.
>
> We consider the potential shortcomings in the answers to the reviewers' questions and hope that they sufficiently address the concerns regarding our experimental section.
>
> ## Questions
>
> > Can you precise why the addition, multiplication, and subtraction in the Bayes Hilbert are well-defined only if we consider $B(\lambda)$ and not $\mathcal{M}(\lambda)$?
>
> Perturbation and powering are well-defined for elements in both $B(\lambda)$  and $ \mathcal{M}(\lambda) $, as these sets consist of measures equivalent to the base measure $\lambda$, ensuring the Radon-Nikodym densities are well-defined.
> In Bayes Hilbert space literature, $B(\lambda)$ is favored because proportional likelihoods, considered equivalent under the likelihood principle, correspond to identification via $=_B$-equivalence (see for example Van den Boogaart et al. [2014].
> This property is not explicitly exploited in our proposed manuscript but added to adhere to the standard framework in Bayes Hilbert space literature.
>
> > You said in the abstract that your framework 'generalizes $f$-divergences by incorporating locally non-convex divergence-generating functions'. This should be clarified as, to my understanding of Section 4, you only consider a certain type of local non-convexity, being functions concave on $[0, a)$ and convex on $[a, \infty)$?
>
> We thank the reviewer for bringing this to our attention. We changed the first sentence in the abstract as follows:
>  > We introduce a novel framework that generalizes $f$-divergences by incorporating divergence-generating functions that exhibit non-convex behavior in a neighborhood of the origin.
>
> > Do you think it is possible to consider functions $f$ with several areas of non-convexity, e.g., a function convex on $[0, a]$, $[b, c]$, $[d, +\infty)$ and concave on $(a, b)$, $(c, d)$?
>
> This idea could be incorporated by adapting the sets $M$ and $N$ in the definition of the mixed conjugate to capture multiple non-convexities.
> To connect $f$-divergences and BHS, we only needed to consider the case where the concave domain is around the origin.
>  If the reviewer thinks we should incorporate the more general cases, i.e., for a function $f:[0,\infty)\to \mathbb{R}$ with $f$ concave on $(A_n)_{n\in\mathbb{N}}\subset [0,\infty), A_i\cap A_j=\emptyset$ for $i\neq j$, we propose adding it to the manuscript.
> We thank the reviewer for highlighting this additional extension.

---

> > ### Author Response · Authors · 2024-11-19
> > **Response to Weaknesses and Questions 2/3**
> >
> > > Experiments: can you please detail the nature of your experiment? It is not clear to understand what you are doing and why you are doing it?
> >
> > Estimating BHS densities or distributions remains a significant challenge due to the functional nature of these spaces, where the observed objects are densities rather than individual samples (see Maier et al. [2021], Birrell et al. [2022]).
> > Methods like kernel density estimation (KDE) must be adapted to operate within the BHS (e.g., it must adhere to a particular closed domain), but KDE struggles with high-dimensional data.
> > Even when a BHS density is successfully estimated, sampling remains difficult.
> > Traditional algorithms, such as Metropolis-Hastings, are computationally infeasible in high dimensions, making them unsuitable for datasets like CIFAR-10, where images have a dimensionality of $32 \cdot 32 \cdot 3 = 3072$.
> >
> > To address this, we train a DCGAN with a $D_{f_{BHS}}$ loss to estimate the density of CIFAR-10 images and generate samples directly from the learned distribution.
> > The GAN's generator maps random vectors to image samples. We draw 100 random vectors, generate corresponding samples, and evaluate them using the FID score.
> > The mean and standard deviation of these scores indicate that our method effectively samples from high-dimensional BHS densities and produces realistic images comparable to those generated by WGANs.
> >
> > We would like to bring the Appendix, where we introduced the underlying algorithm in detail, to the reviewer's attention for specific algorithmic details.
> >
> > We hope this explanation sufficiently addresses the reviewers' questions regarding our experiment.

---

> > > ### Author Response · Authors · 2024-11-19
> > > **Response to Weaknesses and Questions 3/3**
> > >
> > > > [...] could you think of an experiment (even a toy one) with such a heavy-tailed measure where the WGAN would fail and your approach succeed?
> > >
> > > It is not clear whether the distribution of images in the CIFAR-10 dataset is heavy-tailed.
> > >  One simple example based on Yatracos [2022] is learning a lognormal distribution using WGAN
> > > compared to BHSGAN.
> > > Since other reviewers have requested additional experiments, we produced an additional simulation study for this case. In summary, the experiment shows that BHSGAN can approximate the lognormal distribution much better than the Wasserstein GAN.
> > > A detailed description of the experiment can be found below.
> > >
> > > ### Additional Experiments
> > > We train the benchmarking GAN architectures to approximate a heavy-tailed distribution, specifically the lognormal distribution with parameters $(\mathrm{loc}=0, \mathrm{shape}=0.5, \mathrm{scale}=1)$.
> > > The performance of each GAN is assessed by estimating the three parameters and using these estimates to analyze each model's ability to capture the heavy-tailed nature of the distribution.
> > >
> > > From a statistical point of view, the KS test is unsuitable in this context as it does not adequately account for tail behavior.
> > > Therefore, particular attention will be given to comparing the estimated parameter $\mathrm{shape}$ and $\mathrm{scale}$ to evaluate the frameworks' performance in modeling the distribution's tail behavior.
> > >
> > > The steps for this experimental setup can be summarized as follows:
> > >
> > > 1. We first generate 10.000 lognormally distributed samples with parameters $(\mathrm{loc}=0, \mathrm{shape}=0.5, \mathrm{scale}=1)$.
> > > 2. Then, we train each GAN architecture for $35$ epochs using the same set of hyperparameters.
> > > 3. To evaluate the training results, we first generated $1000$ realizations of the estimated distribution for each GAN architecture.
> > >     Then, we estimated the shape and scale parameters for each of the $1000$ realizations and compared standard descriptive statistics such as mean and standard deviation to the true parameters.
> > >
> > > The following table contains the result of this experiment:
> > >
> > > | Name          | shape = 0.5           | scale = 1           |
> > > |---------------|-----------------------|---------------------|
> > > | BHS           | 0.5049 ± 0.0104      | 1.0937 ± 0.0227     |
> > > | Wasserstein   | 0.4220 ± 0.0071      | 1.6670 ± 0.0127     |
> > > | GAN           | 0.1952 ± 0.0028      | 1.3422 ± 0.0130     |
> > > | Pearson       | 0.6181 ± 0.0240      | 0.6863 ± 0.0183     |
> > > | KL            | 0.9951 ± 0.0167      | 1.0126 ± 0.0207     |
> > > | Reverse KL    | 0.8949 ± 0.0178      | 0.7376 ± 0.0120     |
> > >
> > > The BHSGAN approximates the true shape parameter more accurately than every other architecture.
> > > While the KL GAN slightly outperforms the BHSGAN in scale estimation, KL GAN consistently overestimates the shape parameter, with an average value nearly twice the true parameter.
> > > Thus, the BHSGAN emerges as a robust choice for generating heavy-tailed distributions like the lognormal distribution.
> > >
> > > The findings of  the table above can be validated further by considering the newly added figure in the appendix.
> > > BHSGAN approximates the true density well regarding mode and tail behavior.
> > > In comparison, the Wasserstein architecture fails to capture the left tail behavior completely, and the right tail shows some discrepancies in terms of monotonicity.
> > > While the KL GAN captures the true distributions' scale relatively well, the right tail seems to be much heavier than the tail of the true underlying distribution.
> > > Pearson GAN and Reverse KL GAN struggle with oscillations and deviations from the true density, especially in the tails.
> > > The basic GAN performs poorly in terms of alignment, with significant underestimation near the mode and noisy approximation in the tails.
> > >
> > > > I find the nature of your theoretical contribution unclear [...]
> > >
> > > We thank the reviewer for pointing out this inconsistency.
> > >     Upon revising our theory, we propose to adjust the manuscript as follows: Redefining the pseudo-divergence to encompass functions that are concave on an interval $[0, a)$ and convex on $[a, \infty)$, where $a \geq 0$, allows us to include the traditional $f$-divergence as a special case.
> > >     Notably, if $a = 0$, then $[0, a) = \emptyset$, implying $N = \emptyset$.
> > >     Since the sets $M$ and $N$ already encapsulate the convex and concave behaviors, all other definitions and results remain unchanged.
> > >     This minor revision thus establishes a complete connection between $f$-divergences and BHS.
> > >
> > > ## Final remarks
> > >
> > > Again, we thank the reviewer for reading our manuscript carefully and suggesting improvements. We hope that our changes to the manuscript address the reviewer's concerns adequately.

---

> > > > ### Comment · Reviewer_Tns5 · 2024-11-25
> > > > **Thank you**
> > > >
> > > > Thank you for your answer, which solves my main concerns. I am raising my score to 6.

---

### Official Review · Reviewer_zX4e · 2024-10-28

**Soundness:** 3
**Presentation:** 3
**Contribution:** 3
**Rating:** 8
**Confidence:** 4

**Summary:**

The paper proposes a new way to generalize $f$-divergence where the divergence-generating function may not be convex on the whole domain, which they coined as pseudo $f$-divergence. Furthermore, the authors provide a nice connection between the introduced pseudo $f$-divergence and the metric induced by the Bayes Hilbert Space. An illustration of using pseudo $f$-divergence in Bayes Hilbert space GAN demonstrates the potential usefulness of such a generalization.

**Strengths:**

1) The presentation of the submitted manuscript is very clear and its goals are well articulated.

2) Generalization of $f$-divergence and its close connection to Bayes Hilbert Space provides a nice insight as well as practically useful theory to extend existing Bayesian inferences, or probabilistic machine learning tasks.

3) Illustration of introduced methods in GAN further provides soundfulness of the established results.

**Weaknesses:**

No major weakness. Minor weakness below:

1) For the sake of improving the presentation, I would suggest reducing Section 1 and include more details about Bayes Hilbert Space in Section 3—for instance, the paragraph before 1.1 contributions (lines 64-72), the Second paragraph in Bayes Hilbert Spaces part in Section 1(lines 124-131), the second paragraph of section 1.3 (lines 156-160) all substantially overlap. One could summarize three paragraphs into one and just give a high-level depiction in the introduction and provide more details in Section 3 on Bayes Hilbert Space.

2) Many prepositions, or phrases missing in front of references. For instance, in lines 128-129, it is better to write "including" before you list the references. Or perhaps include references inside the brackets [ ].

3) In the Fenchel conjugate definition, I believe it should be x^Ty, not just xy.

**Questions:**

Thanks for the interesting work. Two questions I have are

1) I wonder if the authors could clarify how the pseudo-divergence could yield a new sampling approach at a high level. I understand that these could serve as an alternative approach to measuring the difference between distributions, but it's not clear to me if any sampling procedures can be based on them. Perhaps the more natural way to utilize this result would be variational inference, but I am not sure about the sampling strategies, so any clarification would help.

2) Could authors perhaps include any concrete example where the generalization of $f$-divergence significantly improved or robustified the performance of learning? Or at least, an extensive numerical comparison with the existing divergences to further gain insights (even for simple tasks).

---

> ### Author Response · Authors · 2024-11-19
> **Response to Weaknesses and Questions 1/2**
>
> Dear reviewer,
>
> first of all, thank you for your positive feedback, for reading the manuscript carefully, and for suggesting improvements.
> A reply to your questions and suggestions can be found below.
>
> ## Weaknesses
>
> > For the sake of improving the presentation, I would suggest reducing Section 1.
>
> Since Bayes Hilbert spaces are not widely known in the machine learning community, we initially aimed to provide a detailed introduction in Section 1. However, given the substantial overlap in content, we incorporated the following changes based on the provided suggestions:
>
> - Keep lines 64-72 in Section 1 to motivate the connection between $f$-divergences and the BHS-divergence.
> - Reduce lines 124-131 and therefore remove overlap by replacing lines 124-128 with the following sentence that still provides context to the subsequently listed related works on Bayes Hilbert spaces in terms of functional regression on Bayes spaces:
>  > Bayes Hilbert spaces are frequently used in functional regression, including Maier et al. [2021], [...], where Radon-Nikodym derivatives are modeled using spline functions within the Bayes Hilbert space.
> - To maintain the structure of our outline and reduce substantial overlap with lines 48-53 and 64-72, we replaced lines 156-160 with the following paragraph, which summarizes Section 3:
>  > In Section 3, we review Bayes Hilbert spaces as a framework for measuring distances between probability measures by embedding them into function spaces with a separable Hilbert space structure defined through the clr. The induced metric corresponds to the log-variance, which connects to $f$-divergences.
>
> > Many prepositions, or phrases missing in front of references.
>
> We apologize for submitting an unpolished manuscript regarding citation style. Therefore, as kindly suggested, we added prepositions and used the `\citep` command to add parentheses to references to improve readability.
>
> > In the Fenchel conjugate definition, I believe it should be $x^Ty$, not just $xy$.
>
> In the most general definition of the Fenchel conjugate, the notation $x^Ty$ is indeed used.
> Since our setting only entails real-valued variables and functions, we omitted this notation.
> However, to use notations that are consistent with the literature, we revised the respective definitions in lines 187 and 208 and thank the reviewer for bringing this to our attention.
>
> ## Questions
>
> > I wonder if the authors could clarify how the pseudo-divergence could yield a new sampling approach at a high level.
>
> The pseudo-divergence itself does introduce a new sampling method for Bayes Hilbert spaces.
>         However, by incorporating it into Generative Adversarial Networks (GANs) or other generative models, we can use them for sampling in Bayes Hilbert spaces.
>         In the GAN setting, we train a deep convolutional GAN with $D_{f_{BHS}}$ loss and sample new images from the learned distribution using the generator of the GAN network.
>         To sample new images from the generator, the GAN generator is passed a random vector and returns an image sample.
>         While image domains are rarely discussed in the Bayes Hilbert space literature, they are prevalent in machine learning, which motivates our application of pseudo-divergences to GANs in the context of images.
>
> > Perhaps the more natural way to utilize this result would be variational inference,
>
> Could the reviewer please elaborate on the variational inference approach they have in mind?
> The proposed algorithm in our article utilizes a generalized variational estimation framework
> established by Nguyen et al. [2010].

---

> > ### Author Response · Authors · 2024-11-19
> > **Response to Weaknesses and Questions 2/2**
> >
> > > Could authors perhaps include any concrete example where the generalization of $f$-divergence significantly improved or robustified the performance of learning?
> >
> > As pointed out by the reviewer, our experiments leave room for improvement.
> > We therefore conducted an additional experiment where the BHSGAN architecture approximates a heavy-tailed distribution better than the other GAN architectures. A summary of this experiment can be found below. Details and Figures have been included in the newly proposed manuscript.
> >
> > ### Additional Experiments
> > We train the benchmarking GAN architectures to approximate a heavy-tailed distribution, specifically the lognormal distribution with parameters $(\mathrm{loc}=0, \mathrm{shape}=0.5, \mathrm{scale}=1)$.
> > The performance of each GAN is assessed by estimating the three parameters and using these estimates to analyze each model's ability to capture the heavy-tailed nature of the distribution.
> >
> > From a statistical point of view, the KS test is unsuitable in this context as it does not adequately account for tail behavior.
> > Therefore, particular attention will be given to comparing the estimated parameter $\mathrm{shape}$ and $\mathrm{scale}$ to evaluate the frameworks' performance in modeling the distribution's tail behavior.
> >
> > The steps for this experimental setup can be summarized as follows:
> >
> > 1. We first generate 10.000 lognormally distributed samples with parameters $(\mathrm{loc}=0, \mathrm{shape}=0.5, \mathrm{scale}=1)$.
> > 2. Then, we train each GAN architecture for $35$ epochs using the same set of hyperparameters.
> > 3. To evaluate the training results, we first generated $1000$ realizations of the estimated distribution for each GAN architecture.
> >     Then, we estimated the shape and scale parameters for each of the $1000$ realizations and compared standard descriptive statistics such as mean and standard deviation to the true parameters.
> >
> > The following table contains the result of this experiment:
> >
> > | Name          | shape = 0.5           | scale = 1           |
> > |---------------|-----------------------|---------------------|
> > | BHS           | 0.5049 ± 0.0104      | 1.0937 ± 0.0227     |
> > | Wasserstein   | 0.4220 ± 0.0071      | 1.6670 ± 0.0127     |
> > | GAN           | 0.1952 ± 0.0028      | 1.3422 ± 0.0130     |
> > | Pearson       | 0.6181 ± 0.0240      | 0.6863 ± 0.0183     |
> > | KL            | 0.9951 ± 0.0167      | 1.0126 ± 0.0207     |
> > | Reverse KL    | 0.8949 ± 0.0178      | 0.7376 ± 0.0120     |
> >
> > The BHSGAN approximates the true shape parameter more accurately than every other architecture.
> > While the KL GAN slightly outperforms the BHSGAN in scale estimation, KL GAN consistently overestimates the shape parameter, with an average value nearly twice the true parameter.
> > Thus, the BHSGAN emerges as a robust choice for generating heavy-tailed distributions like the lognormal distribution.
> >
> > The findings of  the table above can be validated further by considering the newly added figure in the appendix.
> > BHSGAN approximates the true density well regarding mode and tail behavior.
> > In comparison, the Wasserstein architecture fails to capture the left tail behavior completely, and the right tail shows some discrepancies in terms of monotonicity.
> > While the KL GAN captures the true distributions' scale relatively well, the right tail seems to be much heavier than the tail of the true underlying distribution.
> > Pearson GAN and Reverse KL GAN struggle with oscillations and deviations from the true density, especially in the tails.
> > The basic GAN performs poorly in terms of alignment, with significant underestimation near the mode and noisy approximation in the tails.
> >
> > ## Final Remarks
> >
> > Again, we thank the reviewer for their positive feedback and detailed suggestions to further improve our manuscript.
> > We would happily engage in further discussions if the reviewer has additional questions or concerns.

---

### Official Review · Reviewer_ScwD · 2024-11-01

**Soundness:** 3
**Presentation:** 2
**Contribution:** 2
**Rating:** 6
**Confidence:** 3

**Summary:**

The paper studies a more general class of f-divergences that does not need to be convex on the whole domain and can be concave on the interval of $[0, a)$ and convex on $[a, \infty)$, called pseudo-divergence-generating function. The authors also link the pseudo f-divergence with the metric induced by the Bayes Hilbert space. The paper uses the proposed framework to extend f-GANs to include their locally non-convex divergence function, showing comparable results with older methods.

**Strengths:**

The idea of generalizing the convex divergence functions to locally non-convex is interesting and is a step towards a more general family of divergences. The paper is well-motivated and the connection to Bayes Hilbert space (BHS) is neat. I enjoyed this connection as it helps with the challenge of estimating the distributions using some samples (something that BHS is not good at while the f-divergences use optimization tools and are more flexible).

**Weaknesses:**

While the idea is interesting, the experimental setup does not show a big advancement in having locally concave functions, and one can see that WGAN is still outperforming the proposed method. Did the authors try to have some synthetic example that clearly shows the benefit of the pseudo f-divergence family?

On another note, I think the paper's writing could improve, particularly some details could help the reader. For example:
- Line 201, Equation 1 refers to a wrong equation, I believe the authors wanted to point to the equation line 196.
- Line 221, is the constant c, dependent on A? If yes, maybe the authors could specify that by writing $c_A$.
- Jumping from Eq. 2 to Corollary 4.4's result could seem surprising as it seems to be a jump from sup and inf to only sup. Checking the appendix, I believe explaining the relation between $T$ and $\tilde{T}$ would clarify this.
- Double Equation in line 431
- The images of Fig 2(a) are barely distinguishable. It would help if the authors could change the size of the image.

**Questions:**

The authors say that their framework helps with the high-dimensional sampling for Bayes Hilbert space distributions. I am not sure if I could follow how the proposed framework solves this problem. I would appreciate if the authors could elaborate on that.

---

> ### Author Response · Authors · 2024-11-19
> **Response to Weaknesses 1/2**
>
> Dear reviewer,
> first of all, thank you for reading our manuscript carefully and raising an important question.
> A reply to your concerns and questions can be found below.
>
> ## Weaknesses
>
>
> > Did the authors try to have some synthetic example that clearly shows the benefit of the pseudo f-divergence family?
>
> As pointed out by the reviewer, our experiments leave room for improvement.
> We therefore conducted an additional experiment where the BHSGAN architecture approximates a heavy-tailed distribution better than the other GAN architectures. A summary of this experiment can be found below. Details and Figures have been included in the newly proposed manuscript.
>
> ### Additional Experiments
> We train the benchmarking GAN architectures to approximate a heavy-tailed distribution, specifically the lognormal distribution with parameters $(\mathrm{loc}=0, \mathrm{shape}=0.5, \mathrm{scale}=1)$.
> The performance of each GAN is assessed by estimating the three parameters and using these estimates to analyze each model's ability to capture the heavy-tailed nature of the distribution.
>
> From a statistical point of view, the KS test is unsuitable in this context as it does not adequately account for tail behavior.
> Therefore, particular attention will be given to comparing the estimated parameter $\mathrm{shape}$ and $\mathrm{scale}$ to evaluate the frameworks' performance in modeling the distribution's tail behavior.
>
> The steps for this experimental setup can be summarized as follows:
>
> 1. We first generate 10.000 lognormally distributed samples with parameters $(\mathrm{loc}=0, \mathrm{shape}=0.5, \mathrm{scale}=1)$.
> 2. Then, we train each GAN architecture for $35$ epochs using the same set of hyperparameters.
> 3. To evaluate the training results, we first generated $1000$ realizations of the estimated distribution for each GAN architecture.
>     Then, we estimated the shape and scale parameters for each of the $1000$ realizations and compared standard descriptive statistics such as mean and standard deviation to the true parameters.
>
> The following table contains the result of this experiment:
>
> | Name          | shape = 0.5           | scale = 1           |
> |---------------|-----------------------|---------------------|
> | BHS           | 0.5049 ± 0.0104      | 1.0937 ± 0.0227     |
> | Wasserstein   | 0.4220 ± 0.0071      | 1.6670 ± 0.0127     |
> | GAN           | 0.1952 ± 0.0028      | 1.3422 ± 0.0130     |
> | Pearson       | 0.6181 ± 0.0240      | 0.6863 ± 0.0183     |
> | KL            | 0.9951 ± 0.0167      | 1.0126 ± 0.0207     |
> | Reverse KL    | 0.8949 ± 0.0178      | 0.7376 ± 0.0120     |
>
> The BHSGAN approximates the true shape parameter more accurately than every other architecture.
> While the KL GAN slightly outperforms the BHSGAN in scale estimation, KL GAN consistently overestimates the shape parameter, with an average value nearly twice the true parameter.
> Thus, the BHSGAN emerges as a robust choice for generating heavy-tailed distributions like the lognormal distribution.
>
> The findings of  the table above can be validated further by considering the newly added figure in the appendix.
> BHSGAN approximates the true density well regarding mode and tail behavior.
> In comparison, the Wasserstein architecture fails to capture the left tail behavior completely, and the right tail shows some discrepancies in terms of monotonicity.
> While the KL GAN captures the true distributions' scale relatively well, the right tail seems to be much heavier than the tail of the true underlying distribution.
> Pearson GAN and Reverse KL GAN struggle with oscillations and deviations from the true density, especially in the tails.
> The basic GAN performs poorly in terms of alignment, with significant underestimation near the mode and noisy approximation in the tails.
>
> > Line 201, Equation 1 refers to a wrong equation [...].
>
> We removed the `\nonumber` command from the respective equation. Thank you for bringing this to our attention.
>
> > Line 221, is the constant c, dependent on A?
>
> The constant $c$ does not depend on the set $A$ as stated in the definition. There exists a constant $c>0$ such that $\mu(A) = c\nu(A) $ for all $A\in\mathcal{B}$.
> See for example Van den Boogaart et al. [2014] , Equation 1.
>
> > Jumping from Eq. 2 to Corollary 4.4's result could seem surprising [...].
>
> We agree with the reviewer that the connection could be more obvious and requires looking up the proof in the appendix.
> Using the notation $\bar T$ instead of $T$, we intend to emphasize the different candidate functions for the optimization problem.
> We therefore added the following paragraph right before Corollary 4.4.:
>
> The optimization problem in Equation (2) can be transformed to reduce the two objectives into a single equivalent objective with the same optimum.
>      This transformation leverages the property $\inf(\cdot)=-\sup(\cdot)$ and substituting $T$ with $\bar{T} = T(I_{\{T\in C(M^*)\}}-I_{\{T\in C(N^*)\}})$. For details, see Appendix *Proof of Corollary 4.4*.

---

> > ### Author Response · Authors · 2024-11-19
> > **Response to Weaknesses 2/2 and Question**
> >
> > > Double Equation in line 431
> >
> > We removed \verb|Equation| from the manuscript and thank the reviewer for pointing that out.
> >
> > > The images of Fig 2(a) are barely distinguishable. It would help if the authors could change the size of the image.
> >
> > To conserve space in the manuscript, we initially kept the image size in Figure 2(a) small.
> > We now moved Figure 2(b) (MNIST images) to the appendix, as it only allows a visual comparison, while Figure 2(a) (CIFAR10) includes accompanying FID scores. This adjustment also allows for an increase in the image size of Figure 2(a).
> >
> > ## Question
> >
> > > The authors say that their framework helps with the high-dimensional sampling for Bayes Hilbert space distributions. [...]
> >
> > In our manuscript, we demonstrate how to sample from high-dimensional Bayes Hilbert space (BHS)
> > distributions in the context of image generation using GANs. While GANs are commonly used for
> > image generation, our key contribution is integrating the Bayes Hilbert space distance into this
> > framework.
> > Estimating BHS densities or distributions remains a significant challenge due to the functional nature
> > of these spaces, where the observed objects are densities rather than individual samples (see Maier
> > et al. [2021], Birrell et al. [2022]). Methods like kernel density estimation (KDE) must be adapted
> > to operate within the BHS (e.g. must adhere to a certain closed domain), but KDE struggles with
> > high-dimensional data. Even when a BHS density is successfully estimated, sampling from it is
> > difficult. Traditional algorithms, such as Metropolis-Hastings, are computationally infeasible in high
> > dimensions, making them unsuitable for datasets like CIFAR-10, where images have a dimensionality of $32\cdot 32\cdot 3 = 3072$.
> >
> > To address this, we train a DCGAN with a $D_{f_{BHS}}$ loss to estimate the density of CIFAR-10 images and generate samples directly from the learned distribution.
> > The GAN’s generator maps random vectors to image samples. We draw 100 such random vectors, generate corresponding samples, and evaluate them using the FID score.
> > The mean and standard deviation of these scores indicate that our method not only effectively samples from high-dimensional BHS densities but also produces realistic images comparable to those generated by WGANs.
> >
> > ## Final Remarks
> >
> > Again, we thank the reviewer for reading our manuscript carefully and suggesting improvements.
> > We hope that our changes to the manuscript address the reviewer's concerns adequately.

---

> > > ### Comment · Reviewer_ScwD · 2024-11-25
> > > **Reply to official comments by the authors**
> > >
> > > I thank the authors for their responses to my questions and for adding more experiments and clarifications. The newly added heavy-tailed experiment is a valuable addition. However, I am curious about its reliability and stability when varying the scale and shape parameters. For instance, is it consistently better at estimating scale across a broader range of settings? Exploring this in a wider parameter space could strengthen the results and address any concerns about potential selection bias.

---

> > > > ### Author Response · Authors · 2024-11-27
> > > >
> > > > We appreciate the reviewer's recognition of our responses and additional experiments. We expanded our simulation study to address concerns about selection bias, reliability, and stability. Instead of simulating a single lognormal distribution with $\mathrm{shape} = 0.5$ and $\mathrm{scale} = 1$, we simulated nine combinations of shape and scale parameters using $\mathrm{shape} = (0.5, 0.75, 1.0)$ and $\mathrm{scale} = (1.0, 1.5, 2.0)$, keeping the location parameter fixed at $0$.
> > > >
> > > > A summary of our findings follows, with detailed results in subsequent tables.
> > > >
> > > > In eight experiments, BHSGAN achieved the best estimate for either the shape, scale, or both parameters, demonstrating its effectiveness in estimating heavy-tailed distributions. Notably, in four experiments, BHSGAN outperformed by simultaneously providing the most accurate estimates for both parameters.
> > > > Even when not yielding the best estimate, BHSGAN's results remained close to the true values.
> > > > By contrast, no other GAN architecture consistently delivered reliable estimates for shape and scale parameters.
> > > >
> > > > We hope this addresses the reviewers concerns!
> > > >
> > > > ## Detailed results
> > > >
> > > > | **Name**| `shape = 0.5`| `scale = 1`|
> > > > |--|--|--|
> > > > | BHS| **0.5049 ± 0.0104**    | 1.0937 ± 0.0227       |
> > > > | Wasserstein      | 0.4220 ± 0.0071        | 1.6670 ± 0.0127       |
> > > > | GAN| 0.1952 ± 0.0028        | 1.3422 ± 0.0130       |
> > > > | Pearson | 0.6181 ± 0.0240        | 0.6863 ± 0.0183       |
> > > > | KL| 0.9951 ± 0.0167        | **1.0126 ± 0.0207**   |
> > > > | Reverse KL | 0.8949 ± 0.0178        | 0.7376 ± 0.0120       |
> > > >
> > > > | **Name**| `shape = 0.5`| `scale = 1.5`|
> > > > |--|--|--|
> > > > | BHS| **0.5219 ± 0.0093**    | **1.4439 ± 0.0237**   |
> > > > | Wasserstein | 0.6087 ± 0.0110        | 1.4369 ± 0.0234       |
> > > > | GAN| 0.2134 ± 0.0074        | 1.5988 ± 0.0570       |
> > > > | Pearson | 0.4663 ± 0.0111        | 1.2221 ± 0.0245       |
> > > > | KL| 0.6225 ± 0.0115        | 1.1645 ± 0.0238       |
> > > > | Reverse KL| 0.3661 ± 0.0089        | 1.9905 ± 0.0460       |
> > > >
> > > > | **Name**| `shape = 0.5`| `scale = 2.0`|
> > > > |--|--|--|
> > > > | BHS| 0.5869 ± 0.0090        | **1.7427 ± 0.0245**   |
> > > > | Wasserstein | 0.7038 ± 0.0253        | 0.7458 ± 0.0293       |
> > > > | GAN | 0.5769 ± 0.0089    | 1.7405 ± 0.0235       |
> > > > | Pearson | 0.7465 ± 0.0100        | 1.2572 ± 0.0171       |
> > > > | KL | 1.1199 ± 0.0149        | 0.7195 ± 0.0125       |
> > > > | Reverse KL  |  **0.4418 ± 0.0096**        | 2.2609 ± 0.0494       |
> > > >
> > > > | **Name**| `shape = 0.75`| `scale = 1.0` |
> > > > |--|--|--|
> > > > | BHS| **0.7414 ± 0.0088**    | **0.9524 ± 0.0125**   |
> > > > | Wasserstein| 1.1793 ± 0.0098        | 0.4213 ± 0.0056       |
> > > > | GAN | 0.8274 ± 0.0145        | 0.9104 ± 0.0174       |
> > > > | Pearson| 0.7989 ± 0.0100        | 1.0592 ± 0.0125       |
> > > > | KL | 0.6903 ± 0.0095        | 1.1085 ± 0.0150       |
> > > > | Reverse KL | 0.8021 ± 0.0137        | 0.8918 ± 0.0168       |
> > > >
> > > > | **Name** | `shape = 0.75`| `scale = 1.5`|
> > > > |--|--|--|
> > > > | BHS | 0.7396 ± 0.0108        | **1.5621 ± 0.0217**   |
> > > > | Wasserstein | 1.6082 ± 0.0185        | 2.0938 ± 0.0415       |
> > > > | GAN| 0.8022 ± 0.0122        | 1.3266 ± 0.0210       |
> > > > | Pearson| **0.7689 ± 0.0099**    | 1.7021 ± 0.0250       |
> > > > | KL | 0.7352 ± 0.0095        | 1.3767 ± 0.0179       |
> > > > | Reverse KL | 0.7127 ± 0.0123        | 1.7372 ± 0.0286       |
> > > >
> > > > | **Name** | `shape = 0.75`| `scale = 2.0` |
> > > > |--|--|--|
> > > > | BHS | **0.7953 ± 0.0100**    | **1.8674 ± 0.0237**   |
> > > > | Wasserstein| 0.6275 ± 0.0116        | 1.7831 ± 0.0302       |
> > > > | GAN  | 0.8067 ± 0.0147        | 1.8523 ± 0.0379       |
> > > > | Pearson | 0.6587 ± 0.0105        | 2.3042 ± 0.0311       |
> > > > | KL | NaN                    | NaN                   |
> > > > | Reverse KL | 2.1734 ± 0.0616        | 0.3555 ± 0.0147       |
> > > >
> > > > | **Name** | `shape = 1.0` | `scale = 1.0` |
> > > > |--|--|--|
> > > > | BHS| 1.0037 ± 0.0108        | **0.9623 ± 0.0130**   |
> > > > | Wasserstein | 2.7592 ± 0.0186        | 5.6620 ± 0.1547       |
> > > > | GAN| **1.0008 ± 0.0080**    | 0.9425 ± 0.0115       |
> > > > | Pearson| 0.8833 ± 0.0109        | 1.1809 ± 0.0168       |
> > > > | KL| 0.9610 ± 0.0108        | 1.0767 ± 0.0148       |
> > > > | Reverse KL| 0.7731 ± 0.0102        | 0.9696 ± 0.0136       |
> > > >
> > > > | **Name** | `shape = 1.0`| `scale = 1.5`|
> > > > |--|--|--|
> > > > | BHS| **1.0592 ± 0.0114**    | **1.4144 ± 0.0198**   |
> > > > | Wasserstein | 1.0854 ± 0.0282        | 0.6397 ± 0.0209       |
> > > > | GAN| 0.7805 ± 0.0088        | 0.9846 ± 0.0112       |
> > > > | Pearson | 0.8369 ± 0.0101        | 0.8934 ± 0.0119       |
> > > > | KL | 1.1719 ± 0.0099        | 0.8939 ± 0.0115       |
> > > > | Reverse KL       | 0.9387 ± 0.0114        | 1.0846 ± 0.0165       |
> > > >
> > > > | **Name**| `shape = 1.0`| `scale = 2.0`|
> > > > |--|--|--|
> > > > | BHS | 1.0650 ± 0.0123        | 1.8252 ± 0.0270   |
> > > > | Wasserstein   | 1.3076 ± 0.0216    | 1.4598 ± 0.0347       |
> > > > | GAN   | NaN        | NaN       |
> > > > | Pearson | 1.6151 ± 0.0203        | 3.3599 ± 0.0782       |
> > > > | KL | 1.0908 ± 0.0173        | 1.8041 ± 0.0353       |
> > > > | Reverse KL | **1.0134 ± 0.0124**        | **2.0225 ± 0.0315**      |

---

### Author Response · Authors · 2024-12-03
**General Response by Authors**

We thank the reviewers for carefully reading our paper, providing positive feedback, and suggesting minor revisions that improved our paper's presentation and soundness.
In the following paragraphs, we summarize revisions of the individual reviewers and the changes we subsequently incorporated into our manuscript.

## Experimental Section

Reviewers *ScwD, zX4e, Tns5,* and *Y69u* all pointed out that the experiments showcasing applications of the newly proposed $BHS$-divergence should be improved by providing additional experiments.
(Pseudo) $f$-divergences in a GAN context outperform Wasserstein GANs in approximating heavy-tailed distributions.
We, therefore, enriched our experimental Section by comparing different GAN architectures in terms of their ability to generate a lognormal distribution with parameters $\mathrm{shape}=0.5$ and $\mathrm{scale}=1.0$.
To evaluate the performance, we estimated the shape and scale parameters of the generated distributions and compared them to the true parameter pair.
BHSGAN provides the best estimate for the shape parameter and a close-to-optimal estimate for the scale parameter.

Reviewer *ScwD* subsequently asked us to test multiple parameter combinations to rule out a potential selection bias.
Therefore, we trained each architecture for nine different pairs of shape and scale parameters and compared the estimated parameters of the generated distributions to the true shape and scale parameters.
In this setting, BHSGAN performed best in most experiments by consistently providing estimates close to the true parameter values.

## Presentation

Reviewer *ScwD* pointed out that the reformulation of the optimization problem in Corollary 4.4. is surprising and requires checking the Appendix to explain the underlying transformation.
We, therefore, added some context in a paragraph preceding the Corollary to motivate this reformulation.
Reviewer *ScwD* also suggested increasing the size of Figure 2(a), which we incorporated by moving the generated MNIST digits (Figure 2(b)) to the Appendix and increasing the size of the generated CIFAR10 images (Figure 2(a)).

Reviewer *zX4e* suggested improving the presentation by reducing Section 1, which contained overlapping information regarding Bayes Hilbert spaces.
We revised our manuscript by removing redundant information and restructuring Section 1 so that Bayes Hilbert spaces are introduced more thoroughly in Section 3.

As requested by Reviewer *Y69u*, we adjusted the wording regarding the experimental result to a more nuanced tone.

## Theoretical Results

Reviewer *Tns5* pointed out that our proposed theoretical divergence framework does not include traditional $f$-divergences since we initially only considered divergence-generating functions $f$ that are convex on $[a,\infty)$ with $a>0$.
We adapted the definition of pseudo divergences to include the case $a=0$, which did not lead to any changes in our main contribution.

Reviewer *zX4e* pointed out that our definition of the convex and concave conjugate should be adjusted by replacing $xy$ with $x^Ty$ to adhere to standard literature notation.
We adapted the definitions to adhere to these conventions.

## Closing Remarks

We sincerely thank the reviewers for their insightful feedback, which has significantly strengthened our manuscript.
We hope the revisions and discussions have addressed their questions and resolved any potential concerns about our results.

---

### Meta-Review · Area_Chair_zpcY · 2024-12-19

**Metareview:**

This paper proposes pseudo f-divergence, an extension of the traditional concept of f-divergence, and introduces a novel estimation algorithm based on it. Specifically, the study extends the conventional f-divergence, which relied solely on convex functions, to pseudo f-divergence, which incorporates combinations of convex and concave functions. Furthermore, the paper reveals a fundamental connection between such divergences and the Bayes Hilbert space for certain choices of f.　The estimation method is based on GANs, and the proposed approach demonstrates performance comparable to Wasserstein GANs on several tasks. While many reviewers acknowledged the importance of the paper’s mathematical contributions, they also pointed out insufficient discussion regarding the practical utility of the method and the implications of its relationship with the Bayes Hilbert space.　Despite these shortcomings, the paper introduces a significant new direction for f-divergence in the context of machine learning, making it a valuable contribution to the community. Therefore, I recommend acceptance.

**Additional Comments On Reviewer Discussion:**

All reviewers raised concerns regarding the conditions under which the proposed method demonstrates its effectiveness. In response, the authors conducted additional experiments on toy datasets to address these concerns. However, the experimental settings remain highly limited and lack comprehensiveness. Reviewer Tns5 and Reviewer Y69u requested further insights into the connection between pseudo f-divergence and the Bayes Hilbert space. While the authors provided additional discussions and included some points in the conclusion section, the implications of this connection and its more direct relationship with Bayesian methods remain unclear. Despite these shortcomings in the discussions, the mathematical ideas presented are novel and significant, and they represent an important contribution.

---

### Decision · Program_Chairs · 2025-01-22

Accept (Poster)